# Structural basis of tRNA recognition by the widespread OB fold

Aline Umuhire Juru[1], Rodolfo Ghirlando [1] & Jinwei Zhang [1] ✉

The widespread oligonucleotide/oligosaccharide-binding (OB)-fold recognizes diverse substrates from sugars to nucleic acids and proteins, and plays key roles in genome maintenance, transcription, translation, and tRNA metabolism. OB-containing bacterial Trbp and yeast Arc1p proteins are thought to recognize the tRNA elbow or anticodon regions. Here we report a 2.6 Å co-crystal structure of *Aquifex aeolicus* Trbp111 bound to tRNA[Ile], which reveals that Trbp recognizes tRNAs solely by capturing their 3′ ends. Structural, mutational, and biophysical analyses show that the Trbp/EMAPII-like OB fold precisely recognizes the single-stranded structure, 3′ terminal location, and specific sequence of the 3′ CA dinucleotide − a universal feature of mature tRNAs. Arc1p supplements its OB − tRNA 3′ end interaction with additional contacts that involve an adjacent basic region and the tRNA body. This study uncovers a previously unrecognized mode of tRNA recognition by an ancient protein fold, and provides insights into protein-mediated tRNA aminoacylation, folding, localization, trafficking, and piracy.

tRNAs are among the most abundant and essential biomolecules in the cell. Besides decoding the genetic code on the ribosome, tRNAs and their fragments also perform key regulatory functions in transcriptional and translational control, stress responses, and antiviral immunity[1]. They are further exploited by retroviruses such as HIV-1 to serve as obligatory viral replication primers and regulators of virion assembly[2–5]. All known structural and sequence features of the tRNA are recognized by a wide array of protein and RNA machineries that drive tRNA biogenesis, processing, folding, modification, localization, trafficking, aminoacylation, translation, and decay[6,7]. Incorrect or deregulated tRNA recognition leads to numerous disease states, including neurological disorders and cancers[8,9]. The ever-expanding tRNA interactome employs diverse protein folds to recognize the tRNA structure, including the double-stranded RNA-binding motif (dsRBM)[10], helix-turn-helix (HTH) motif[11], oligonucleotide/oligo-saccharide-binding (OB) fold[12,13], pentatricopeptide repeat (PPR) domain[14], pseudouridine synthase and archaeosine transglycosylase (PUA) domain[15,16], RNA recognition motif (RRM)[17], Rossman fold[18], thiouridine synthases, RNA methylases and pseudouridine synthases (THUMP) domain[19,20], WHEP[21] and Zinc-finger[22] domains. However, most of these domains alone only exhibit marginal affinities and specificities towards tRNA and must collaborate to achieve biologically

relevant affinities and selectivity via multivalent tRNA binding. By contrast, the OB-fold[12,13], the PPR[14], and several others are capable of general tRNA recognition on their own.

The OB-fold is an ancient protein superfold composed of 5 β-strands assembled into a small barrel and capped by a short α-helix[23,24]. This compact fold is structurally conserved despite relatively low sequence conservation[25]. The OB-fold utilizes the outer surface of the barrel and its protruding loops connecting the β-strands to bind diverse ligands, from proteins to oligosaccharides and various nucleic acid structures, including single-stranded (ss) DNA, ssRNA, and the tRNA anticodon loop. Across all domains of life, a widespread class of OB folds have specialized to recognize and manipulate tRNAs, perform key functions in tRNA trafficking and metabolism, and are designated as the tRNA binding domain (TRBD) superfamily (IPR002547; PF01588).

In bacteria, *Aquifex aeolicus* Trbp111 (tRNA-binding protein, 111 amino acids) and *Escherichia coli* Trbp exemplify the free-standing, dimeric OB-fold proteins that bind diverse tRNA species. Trbp has been suggested to facilitate the folding of the tRNA structure by binding its characteristic elbow region formed by the intercalation of the D-loop into the T-loop and was observed to enhance aminoacylation by several aminoacyl-tRNA synthetases (aaRSs)[12,26]. Interestingly,

[1]Laboratory of Molecular Biology, National Institute of Diabetes and Digestive and Kidney Diseases, Bethesda, MD, USA. ✉e-mail: jinwei.zhang@nih.gov

some bacterial species that lack Trbp, such as *Thermus thermophilus*, contain a homologous CsaA protein, which shares ~38% sequence identity with Trbp and a highly similar structure (~1.8 Å RMSD). CsaA is a protein secretion chaperone reported to also bind tRNAs[27]. In yeast, the aminoacyl-tRNA synthetase cofactor 1 protein (Arc1p) contains an internal OB-fold preceded by a GST-like N-terminal domain (NTD) and a basic middle domain and followed by a C-terminal domain (CTD) that mimics the Trbp dimer interface (Fig. 1a). Arc1p is the organizational hub of the yeast multi-tRNA synthetase complex (MSC) where it directly associates with both the glutamyl-tRNA synthetase (GluRS) and methionyl-tRNA synthetase (MetRS) via its NTD[28]. Both the OB-fold and middle domain contribute to tRNA binding, and together enhance aminoacylation efficiency by decreasing the $K_m$ for tRNAs[29–31].

In protozoa, such as *Plasmodium* spp., the causative agent of malaria, the tRNA-import protein (tRip) uses a membrane-anchored OB-fold to import human tRNAs into the parasite, to either supplement the tRNA pool in sporozoites for translation or to serve signaling functions[32]. As a result, deletion of tRip impairs the infectivity and fitness of *Plasmodium* sporozoites[32]. In metazoans, the Arc1p role is performed by the aminoacyl-tRNA synthetase complex-interacting multifunctional protein 1 (AIMP1, also known as p43). AIMP1, along with AIMP2 and AIMP3, orchestrate the assembly of the larger metazoan MSCs using its N-terminal extended α-helical domain and recruits tRNAs using its OB-fold[33]. Both domains are required for robust tRNA binding. At the onset of apoptosis, this OB-fold is proteolytically cleaved by caspase 7 to form the endothelial monocyte-activating polypeptide II (EMAPII) cytokine. EMAPII is released from the MSC and secreted to recruit macrophages to engulf apoptotic cells and regulate inflammation and angiogenesis[34,35]. Given that the tRNA-binding ability is lost when the EMAPII domain is cleaved and released from the MSC, AIMP1 serves dual functionalities: (1) gating the entry of tRNAs to the MSC to control translation; (2) harboring the latent EMAPII cytokine to regulate apoptosis[36].

Besides the non-enzymatic Trbp, Arc1p, tRip, and AIMP1/EMAPII proteins, the tRNA-binding OB-fold is frequently found as direct fusions with numerous aaRSs, including bacterial, plant, and nematode

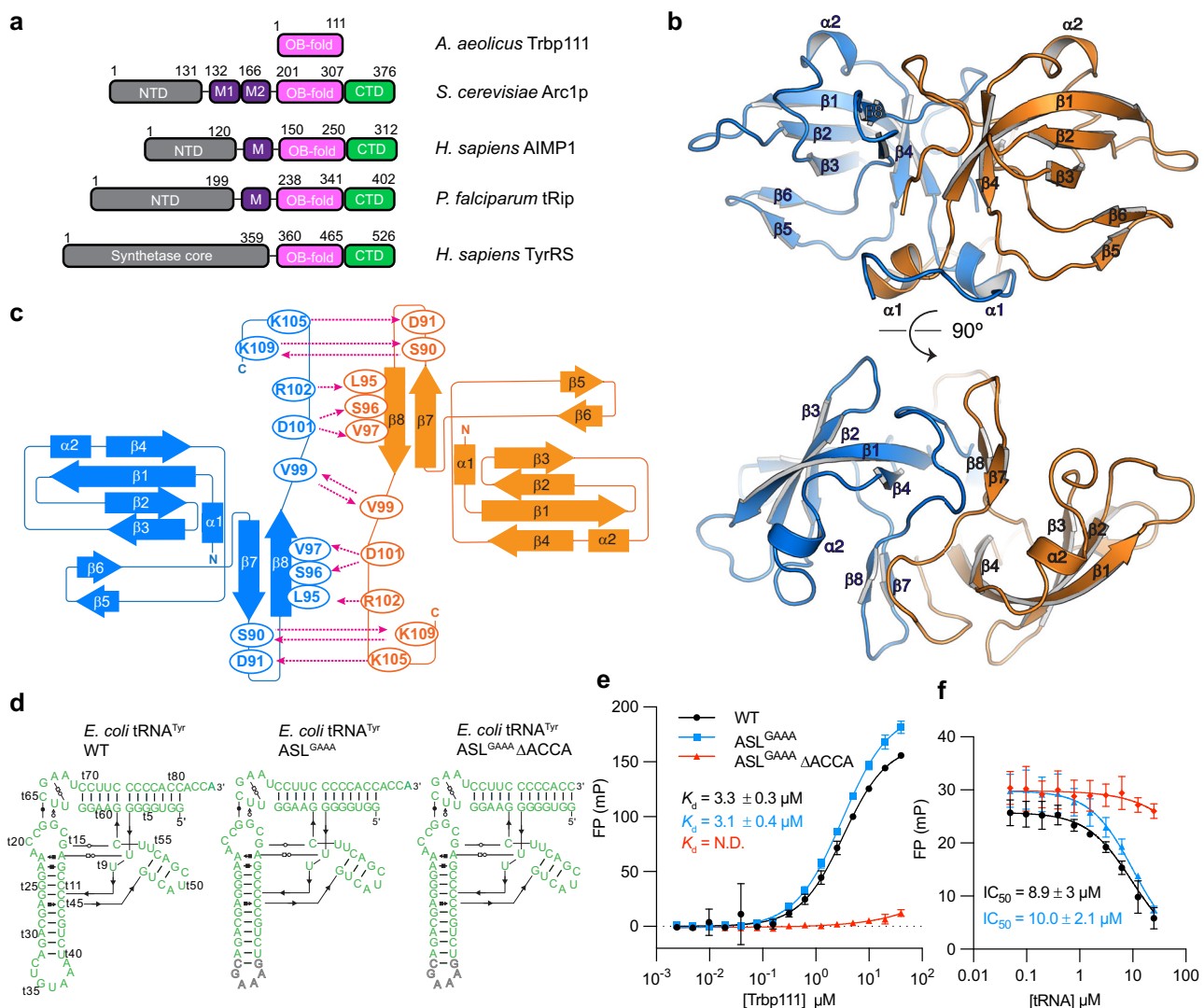

**Fig. 1 | Structure and tRNA-binding of Trbp111. a** Domain organization of Trbp111 and related OB-fold containing proteins. AIMP1: aminoacyl-tRNA synthetase complex-interacting multifunctional protein 1, tRip: tRNA-import protein, TyrRS: Tyrosyl-tRNA synthetase. "M" denotes the middle domain. M1 and M2 are two sub-regions of Arc1p middle domain of similar sizes. **b** Crystal structure of Trbp111 dimer. Monomers are related by a two-fold symmetry. **c** Secondary structure schematic illustration of the bottom view in (**b**). Arrows indicate inter-protomer interactions at the dimeric interface. **d** Secondary structure of *E. coli* tRNA[Tyr] constructs used for Trbp111 binding. "t" prefix denotes tRNA numbering. **e** Fluorescence polarization (FP) assay where Trbp111 was titrated into fluorescein-labeled tRNA. **f** Competition FP assay of tRNAs used in (**e**). Experiments in **e**, **f** were conducted in 25 mM Tris-HCl pH 8, 50 mM KCl, 2 mM MgCl₂. Data are mean ± s.d. n = 3 biologically independent replicates.

MetRSs, *T. thermophilus* PheRS, and human TyrRS (Fig. 1a)[37–42]. Such fusions in unicellular human pathogens such as *Entamoeba histolytica* have been shown to increase aminoacylation efficiency and exacerbate the morbidity caused by these pathogens[37,43]. Similarly, artificially fusing the Arc1p OB domain with a non-tRNA binding peptide synthetase PheA conferred tRNA aminoacylation using an adenylated phenylalanine[44], and another OB fusion with a catalytic MetRS core enhanced its aminoacylation activity[45].

Despite the nearly universal occurrence and functional importance of the Trbp/EMAPII-like OB-fold to numerous facets of tRNA metabolism, where and how it recognizes tRNA have remained unknown. A prevailing proposal based on molecular docking and footprinting analyses suggests that the OB-fold recognizes the elbow region of the tRNA, which would thus leave the amino acid acceptor stem, as well as the anticodon, free to interact with aaRSs[12]. However, other lines of evidence suggest that the OB-fold may interact with the anticodon stem, acceptor stem, or even its 3′ end[46]. To distinguish among these possibilities, we conducted structural and biochemical studies to identify which tRNA region is recognized by the OB-fold, how selectivity for tRNA over other nucleic acids is achieved, and how auxiliary protein domains modulate OB-tRNA binding. We report a co-crystal structure of *Aquifex aeolicus* Trbp111 bound to tRNA^Ile, which reveals an unexpected mode of tRNA recognition via its 3′ end. Contrary to previous proposals, Trbp111 does not recognize the global tRNA structure nor its elbow or anticodon but solely binds its universal 3′ terminal single-stranded CA dinucleotide with a strong positional, structural, and sequence preference. Arc1p similarly binds tRNA 3′ end as Trbp and supplements this contact with an additional contact to the tRNA body (defined as the non-tail portion). Together, our data uncover and visualize a previously unrecognized mode of tRNA recognition employed by the widespread OB superfold.

## Results

### Trbp111 employs an extraordinarily robust homodimer interface

The crystal structure of *Aquifex aeolicus* Trbp111 was previously reported at 2.5 Å resolution in space group C2₁ (PDB: 1PYB)[12]. Here we report another crystal structure of Trbp111 at 2.3 Å in space group C222₁ which exhibits substantially improved backbone and sidechain geometry and reduced clashes (Supplementary Fig. 1 and Table. 1). As described, Trbp111 forms a homodimer where each protomer adopts a canonical OB-fold architecture resembling the Greek key motif, in which five β-strands assemble into two three-stranded β-sheets which are joined to form a β-barrel (Fig. 1b, c). The core OB-fold comprises residues 16–89, with the flanking N-terminal (1–15) and C-terminal (90–111) residues of each protomer interacting in an antiparallel manner, forming a central interfacial region. The interface involves an extensive network of hydrogen bonds, salt bridges, hydrophobic interactions, aromatic stacking, and cation-π interactions, altogether burying ~1955 Å² of solvent-accessible surfaces (Fig. 1c). These contacts produce an unusually robust dimer interface (theoretical $\Delta G$ ~ −22.5 kcal/mol, $K_d$ ~ 3.1 × 10⁻¹⁷ M) that resists denaturation by heat, SDS, Urea, or guanidine treatments (Supplementary Fig. 2). Full denaturation was only achieved when combining heat denaturation and 6 M guanidine treatment. This compact, ultra-stable protein-protein interface is consistent with the hyperthermophilic nature of *A. aeolicus,* which grows at 85–95 °C. Its resistance to heat and chemical insults is in line with proposed functions as chaperones that may aid the folding of other proteins (e.g., CsaA) or tRNAs (e.g., Trbp)[27]. Interestingly, this dimer interface is mimicked by monomeric Arc1p and AIMP1 proteins which fuse an OB-containing functional protomer with a truncated structural protomer to form pseudo-dimer interfaces[35]. This interface is also suggested to be primarily responsible for the dimerization of certain MetRS proteins[41,47]. This widely adopted, unusually stable protein-protein interface may serve as an artificial fusion motif that drives

protein homo- or hetero-dimerization or multimerization to create large designer protein assemblies or aid cryo-EM analyses of small proteins.

### Trbp111 primarily recognizes tRNA 3′ ends

Using a fluorescence polarization (FP) assay where Trbp111 is titrated into tRNAs labeled with fluorescein on their 3′ ends, we observed a $K_d$ ~ 3 μM (Fig. 1e), an affinity significantly lower than the ~30 nM $K_d$ previously estimated by gel-shift analysis[12] but on a par with human EMAPII which binds with $K_d$ ~ 20–40 μM[34]. Considering that 3′ end labeling of tRNA may impact its interaction with Trbp111, we also conducted a competition FP assay, where unlabeled tRNA is titrated into a pre-formed Trbp111 complex with labeled tRNA. This assay format using unlabeled tRNA also reported micromolar affinity and showed that the 3′ fluorescein did not drastically impact Trbp111 binding (Fig. 1f). To further corroborate the FP findings, we performed analytical ultracentrifugation (AUC) analyses, which affirmed the micromolar affinity between Trbp111 and tRNAs (Supplementary Fig. 3). The large variation of observed $K_d$s may stem from major differences in the assay formats or conditions. Gel matrices are known to exert caging effects on protein-nucleic acid complexes as they reduce diffusion and promote local reassociation[48]. The 4 °C temperature used in the gel analyses may further stabilize the complex compared to the room temperature FP measurements performed in solution.

As different regions and structural features of the tRNA have been suggested to be recognized by Trbp111, we then sought to define the regions of tRNA that contribute to binding. Replacing the tRNA anticodon stem loop (ASL) with a structurally distinct GAAA tetraloop had no impact on the affinity, suggesting that the ASL region does not contribute to binding (Fig. 1e, f). By contrast, deletion of the 3′ terminal ACCA sequence abolished binding, suggesting that Trbp111 primarily recognizes the tRNA 3′ end as opposed to the tRNA body.

### Trbp111-tRNA co-crystal structure reveals a previously unobserved binding mode

To confirm our in-solution finding that Trbp111 binds the tRNA 3′ end and to visualize the interface, we determined its co-crystal structure at 2.6 Å by engineering the tRNA ASL region which does not contribute to binding. We observed two tRNAs bound to a Trbp111 dimer *in crystallo* employing nearly identical interfaces, each burying ~484 Å² of solvent-accessible surfaces (Fig. 2a and Supplementary Figs. 4, 5). However, sedimentation velocity AUC experiments primarily observed a single tRNA bound to the Trbp111 dimer in solution (Supplementary Fig. 3). This result is congruent with a previous stoichiometry analysis[13], and hints at negative cooperativity of the two tRNA binding sites on the Trbp111 dimer via protein allostery. Notably, the relatively small ~484 Å² tRNA-Trbp111 interface is comparable to the ~500 Å² buried at the tRNA 3′ end-T-box riboswitch discriminator interface, which produces 2 μM affinity[49], consistent with the ~3 μM Trbp111 affinity estimated from FP and AUC analyses.

The structure reveals acute bending of the tRNA terminal dinucleotide (tC75-tA76) away from the tRNA body and its preceding single-stranded nucleotides (tA73, tC74, Fig. 2b, c). Comparing the two complexes in the asymmetric unit, the Trbp111-bound tC75-tA76 dinucleotide pivots relative to the tRNA body as a rigid body due to the flexible tA73-tC74 linker (Supplementary Fig. 6). Interestingly, tC75 and tA76 are independently recognized by two pre-formed sequence-selective binding pockets. The penultimate tC75 nucleobase forms a T-shaped π-π stacking interaction with F78 and is further anchored by hydrogen bonds from the backbone amide nitrogen atoms of E32 and K33 (Fig. 2b–d). The immediately preceding tC74 is oriented away from the protein surface so as not to clash with F78 (Fig. 2b, c). The terminal tA76 is captured by a predominantly hydrophobic pocket, with the nucleobase sandwiched by M85 on one side and the R75 methylene groups and I77 on the other (Fig. 2c, e, f). The tA76

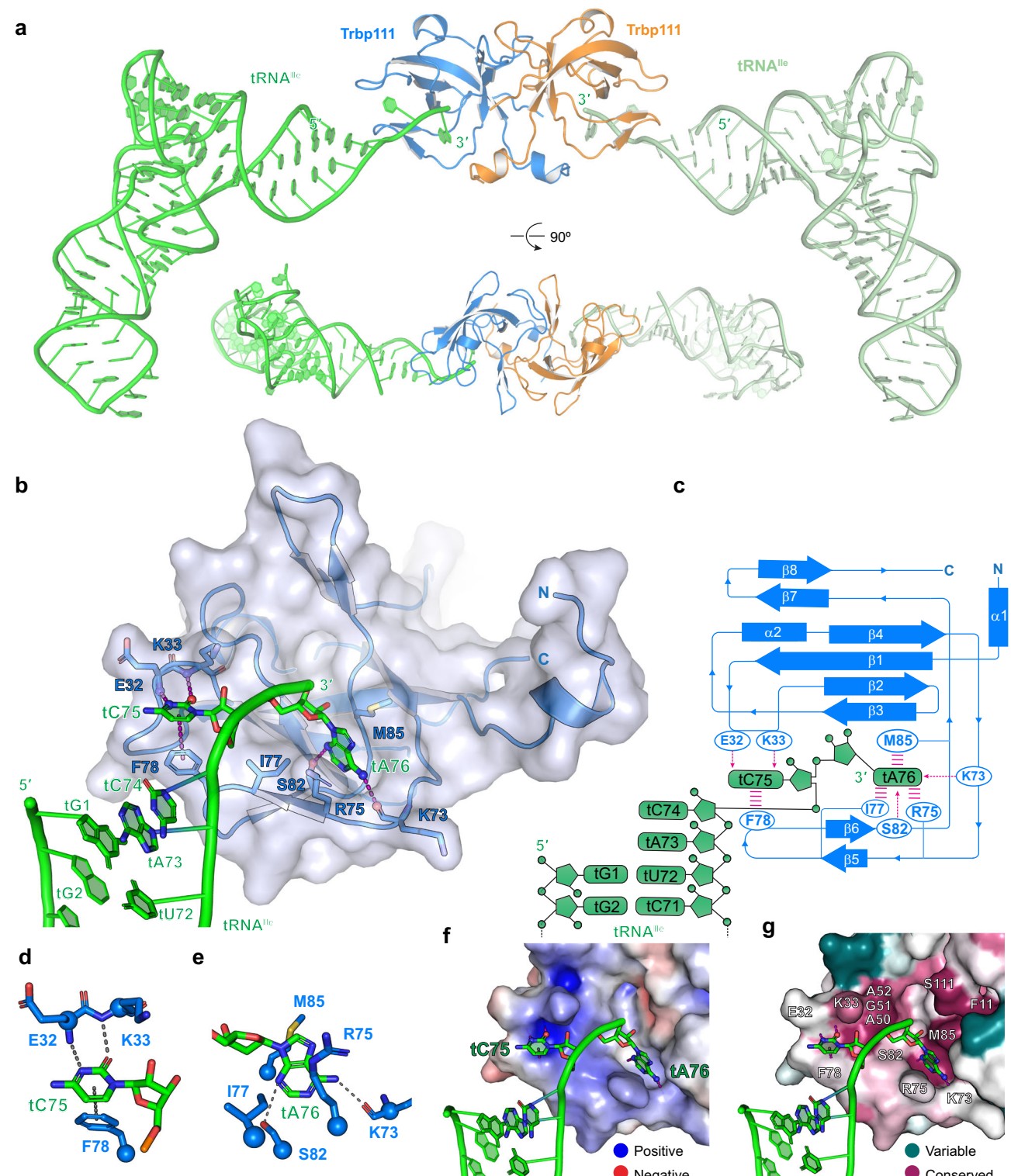

**Fig. 2 | Co-crystal structure of Trbp111 in complex with *E. coli* tRNA^Ile. a** Overall structure of Trbp111 dimer bound to two tRNA molecules. **b** Detailed view of the Trbp111-tRNA interface. **c** Schematic representation of the interface in (**b**). Dashed arrows represent hydrogen bonds; parallel magenta lines represent stacking interactions. **d**, **e** Recognition of tC75 (**d**) and tA76 (**e**) by Trbp111. **f** Electrostatic surface of the tRNA-binding site on Trbp111. Red: negatively charged; blue: positively charged. **g** Conservation analysis of the Trbp111 tRNA-binding site using ConSurf[77]. Dark red: conserved; cyan: variable.

nucleobase is further recognized by hydrogen bonds from S82 and the K73 carbonyl group. Single substitutions of evolutionarily conserved, interfacial Trbp111 residues impaired tRNA binding, with the largest defects from R75A nearly abolishing binding, followed by K33A, S82A, M85A, and F78A (Figs. 2g, 3a, b). Interestingly, the adjacent P74A which

is expected to remove a backbone bend at the base of L$_{45}$ loop (using conventional numbering of OB-fold) had little effect on binding. This suggests that the L$_{45}$ loop tolerates backbone perturbations likely owing to its stable β hairpin structure (β5-β6, Fig. 2c). Overall, our Trbp111 mutational and binding analyses in solution are consistent

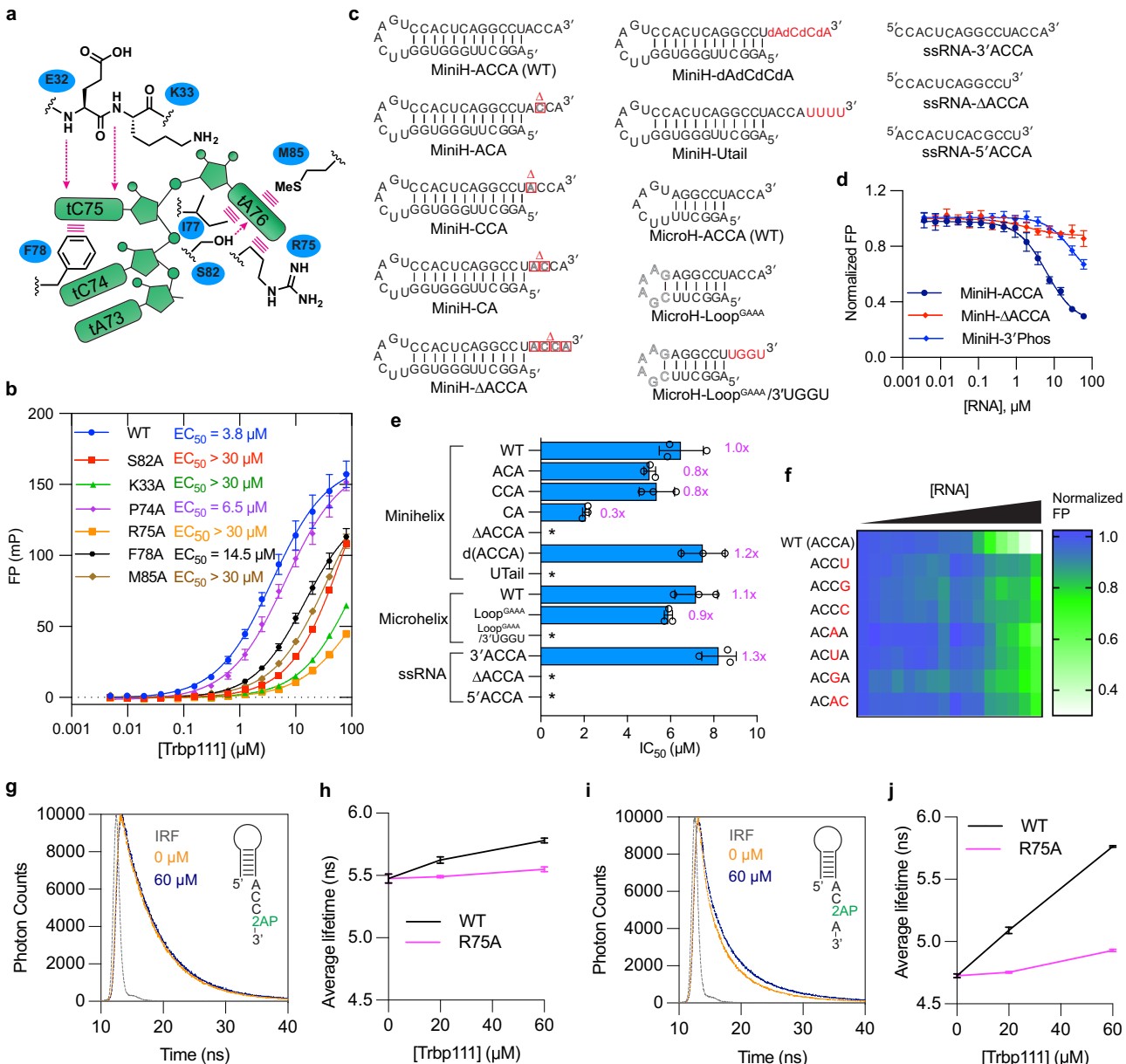

**Fig. 3 | Mutational analysis of the Trbp111-tRNA interface. a** Illustration of Trbp111 chemical moieties contacting tRNA. **b** Fluorescence polarization analysis of WT and mutant Trbp111 binding to fluorescein-labeled *E. coli* tRNA^Tyr. **c** Secondary structures of oligonucleotides used in competition FP assays in (**d–f**). MiniH: minihelix; MicroH: microhelix. **d** Representative titration curves from FP assays showing the effects of 3'ACCA deletion from the minihelix or addition of a 3' phosphate. **e** FP competition $IC_{50}$ values of tested oligos. Asterisks: weak or undetectable binding, where $IC_{50}$ value could not be determined. **f** Heat-map representation of FP values in competition experiments as in (**d**) examining the effects of 3' end base substitutions in minihelices. Mutated positions are indicated in red. **g–j** Effects of Trbp111 on fluorescence lifetime of 2AP at minihelix position tA76 (MiniH-ACC-2AP) or tC75 (MiniH-AC-2AP-A). **g** Representative MiniH-ACC-2AP decay curves in the absence or presence of 60 μM wildtype Trbp111. **h** Average MiniH-ACC-2AP lifetimes in the presence of wildtype or R75A Trbp111. **i** Representative MiniH-AC-2AP-A decay curves in the absence or presence of 60 μM wildtype Trbp111. **j** Average MiniH-AC-2AP-A lifetimes in the presence of wildtype or R75A Trbp111. IRF: instrument response function. Data are mean ± s.d. *n* = 3 biologically independent replicates.

with previous gel-shift analyses[12], extend the previous findings with additional mutants, and place these effects in the structural context.

Our Trbp111 apo structure superposes very well with the complex (RMSD ~ 0.6 Å), with key side chains occupying nearly identical positions (Supplementary Fig. 6). Therefore, Trbp111 is pre-organized to capture the terminal CA dinucleotide of tRNAs. While this binding mode was previously unobserved and largely unexpected, several previous observations lend it strong support. First, aptamers selected in vitro for binding to *Aeropyrum pernix* Trbp converged on possessing a 3' CA dinucleotide overhang, the deletion of which abolished binding[50]. Second, the C-terminal OB-

fold domain of *Nanoarchaeum equitans* MetRS preferentially bound the 3' half of tRNA over the 5' half, and the binding required the tRNA 3' NCCA tail sequence[38]. Finally, the Arc1p OB domain – PheA fusion protein exhibited nearly identical aminoacylation efficiencies towards six different tRNAs, including initiator tRNAs[44], most of which have locally divergent elbow structures. By contrast, the elbow-recognizing HIV-1 Gag Matrix protein bound distinct tRNAs with drastically different affinities due to natural variations in the D-loop[3,4]. Together, these previous findings and our new data suggest Trbp111 binds the tRNA 3' end but not the elbow.

## Trbp111-tRNA binding is precisely localized to the terminal CA nucleobases

To define the sequence and structural specificity of Trbp111 interaction with the tRNA 3′ end, we conducted detailed FP competition assays using a panel of *E. coli* tRNA$^{\text{Ile}}$ minihelices (containing the fused T stem-loop and acceptor stem), microhelices (only the acceptor stem), and short ssRNA oligos (Fig. 3c). Using this simplified system, we first verified that, as is with full-length tRNAs, deletion of the 3′ ACCA tail from the minihelix also abrogated binding (Fig. 3c, d), in full agreement with the co-crystal structure. Then we tested the functional sufficiency of the 3′ CA by deleting the discriminator tA73 or tC74 and found that neither reduced Trbp111 binding (Fig. 3c, e). Curiously, deleting both actually increased the affinity by three-fold, which may reflect a gain of contacts between the duplex terminal base pair and Trbp111. These data suggest that the preceding tA73-tC74 dinucleotide does not contribute to binding, and the terminal CA dinucleotide confers full Trbp111 binding.

The co-crystal structure shows that the Trbp111-bound tRNA 3′ end is partially exposed to the solvent, but its trajectory is directed toward the outer wall of the OB barrel. To test if Trbp111 tolerates significant 3′ extensions and can recognize internal CA nucleotides, we tested binding to a minihelix appended with a UUUU tetranucleotide. This extension abolished binding (Fig. 3e and Supplementary Fig. 7), suggesting that while a fluorescein moiety conjugated to a cleaved ribose is well tolerated, a longer 3′ extension is not. Similarly, the addition of a single phosphate group to the intact terminal ribose significantly reduced binding (Fig. 3d), suggesting that negative charge is not well tolerated by the hydrophobic pocket. We also note the presence of CA dinucleotides in the T-loop that caps the minihelices and microhelices, which do not confer Trbp111 binding. Thus, Trbp111 has evolved to precisely recognize the 3′ terminus of mature tRNAs. We next tested the sequence specificity for the terminal CA dinucleotide. All substitutions at either tC75 or tA76 position significantly impaired binding (Fig. 3f and Supplementary Fig. 7). However, the tC75 substitutions retained more residual binding compared to tA76 mutations, which generally blocked binding. These data suggest a robust sequence specificity achieved by the individual tC75 and tA76 binding sites, with the latter being more functionally critical for the Trbp111 interaction, consistent with the more extensive contacts to tA76 observed in the co-crystal structure (Supplementary Fig. 5). To obtain direct, physical evidence that Trbp111 binds the tRNA 3′ end in solution, we tested the effect of Trbp111 on the fluorescence lifetime of a 2-aminopurine (2AP) nucleobase analog incorporated at either the tA76 or tC75 position in two minihelices. As expected, without proteins, the 2AP located at tC75 is substantially more quenched than at tA76 (4.7 vs 5.5 ns, uncertainties < 1%), consistent with the former being sandwiched by two nucleobases while the latter, terminal nucleotide, is only transiently quenched by its penultimate neighbor through stacking[51,52]. In both cases, the average 2AP lifetimes increased significantly in the presence of wildtype Trbp111 (reaching up to 5.8 ns, Fig. 3g–j, Supplementary Fig. 8 and Table 2), consistent with the complete unstacking of the CA dinucleotide observed in the Trbp111-bound structure. The R75A mutant resulted in a much smaller increase in lifetime and only at the highest concentration tested, consistent with the binding defect observed in the FP assays.

The minihelices partially retain the elbow structure of tRNAs and feature the same T-loop - 3′ end distance. To further ascertain if the elbow/T-loop region contributes to Trbp binding by making a secondary, distal contact, we further shortened the stem to measure binding to the microhelices. Microhelices exhibited approximately the same IC$_{50}$ value ($7.2 \pm 1.0\,\mu$M) as minihelices ($6.5 \pm 1.0\,\mu$M) (Fig. 3e). Substitution of the characteristic T-loop in the microhelices with a structurally distinct GAAA tetraloop did not affect binding. These data suggest that the elbow or the T-loop does not contribute to Trbp111 binding. By contrast, the replacement of the terminal

ACCA sequence with UGGU completely abolished binding, consistent with findings using the full tRNAs and minihelices. Finally, we asked if a ssRNA oligo that terminates with a 3′ ACCA sequence can recapitulate tRNA binding. Remarkably, such an oligo exhibited an IC$_{50}$ value ($8.2 \pm 0.8\,\mu$M) comparable to that of the minihelices and microhelices. Removing the 3′ ACCA sequence or moving it to the 5′ end abolished binding. These data demonstrate that Trbp111 exclusively binds 3′ terminal CA dinucleotides and rejects 5′ terminal or internal CAs.

Another unusual feature of the tRNA recognition mode of Trbp111 is that it does not seem to recognize the ribose hydroxyl groups, including the *cis* diol of tA76. The crystal structure shows the proximity of the side chain of K33 to the tC75 2′ OH ( ∼ 3.5 Å), which may confer preferred binding to RNA. To ask if Trbp111 can also bind DNA tails, we used a chimeric minihelix that bears four terminal deoxyriboses in place of the riboses. The DNA tail exhibited comparable affinity as its RNA counterpart (Fig. 3e), suggesting that the Trbp111 OB-fold is adaptable enough to bind both RNA and DNA tails, and may even accommodate aminoacyl-tRNAs since it does not recognize the *cis* diol and tolerates some 3′ adducts. This result corroborates the structural observation that Trbp111 focuses its recognition on the nucleobases rather than the sugar-phosphate backbone. This unusual approach contrasts with most RNA-binding proteins, including other tRNA 3′-end binding proteins such as aaRSs[53], EF-Tu[54], Exportin-t[55], and CGI121 subunit of the KEOPS tRNA-modifying complex[56], which employ larger interfaces that also involve the RNA backbone (Supplementary Fig. 9).

## Arc1p requires both its middle and OB domains, but not its NTD, to bind tRNAs

To ask whether the observed Trbp111-tRNA binding mode also applies to other tRNA-binding OB folds, we examined yeast Arc1p, consisting of a GST-like NTD, a lysine-rich middle domain, the OB-fold, and a CTD that appears to stabilize the OB-fold via a pseudo-dimer interface. Using isothermal titration calorimetry (ITC) and FP, we observed that the full-length protein has an affinity of $1.30 \pm 0.01\,\mu$M for *E. coli* tRNA$^{\text{Ile}}$ (Fig. 4a, b and Supplementary Table 3). Unexpectedly, deletion of the NTD (residues 1–131) resulted in five-fold enhanced affinity (250–360 nM, Figs. 4a, b, 5a), suggesting an autoinhibition mechanism. Further truncation into the middle domain (ΔNΔM1) reduced binding by ∼ 10-fold while its entire deletion (ΔNΔM) abrogated binding. Removing the OB-fold also abolished binding, suggesting that both the middle domain and the OB-fold are required for tRNA interaction. These findings are congruent with an earlier report that the Arc1p middle domain binds RNA non-specifically[57] and that AIMP1 also requires both an accessory domain and the EMAPII/OB domain for robust tRNA binding (Fig. 1a).

## Arc1p recognizes tRNA 3′ end in a manner similar to Trbp111

The crystal structure of Arc1p OB-fold superposes closely with Trbp111 in our co-crystal structure (RMSD = 1.2 Å), placing its S278, M281, and V271 residues at equivalent positions as the tA76-binding S82, M85 and R75 residues of Trbp111, respectively (Fig. 4c–e). Further, Arc1p contains D226, S227, and R274 in place of Trbp111's tC75-binding E32, K33, and F78, respectively, where the R274 is well positioned to stack with tC75 like F78. Mutating these equivalent residues in Arc1pΔN reduced tRNA binding similarly to Trbp111 by up to 9-fold (R274A, Fig. 4f). Arc1p binding exhibited a reduced dependency on the hydrophobic residues that sandwich tA76. Not only did the M281A substitution cause no defect, even a V271A/M273A/M281A triple substitution only marginally reduced binding. This observation could be partly explained by the similarly hydrophobic alanines maintaining comparable van der Waals contacts to tA76. Alternatively, tRNA binding to Arc1p may be more resilient due to bivalent binding to tRNA from its middle and OB domains.

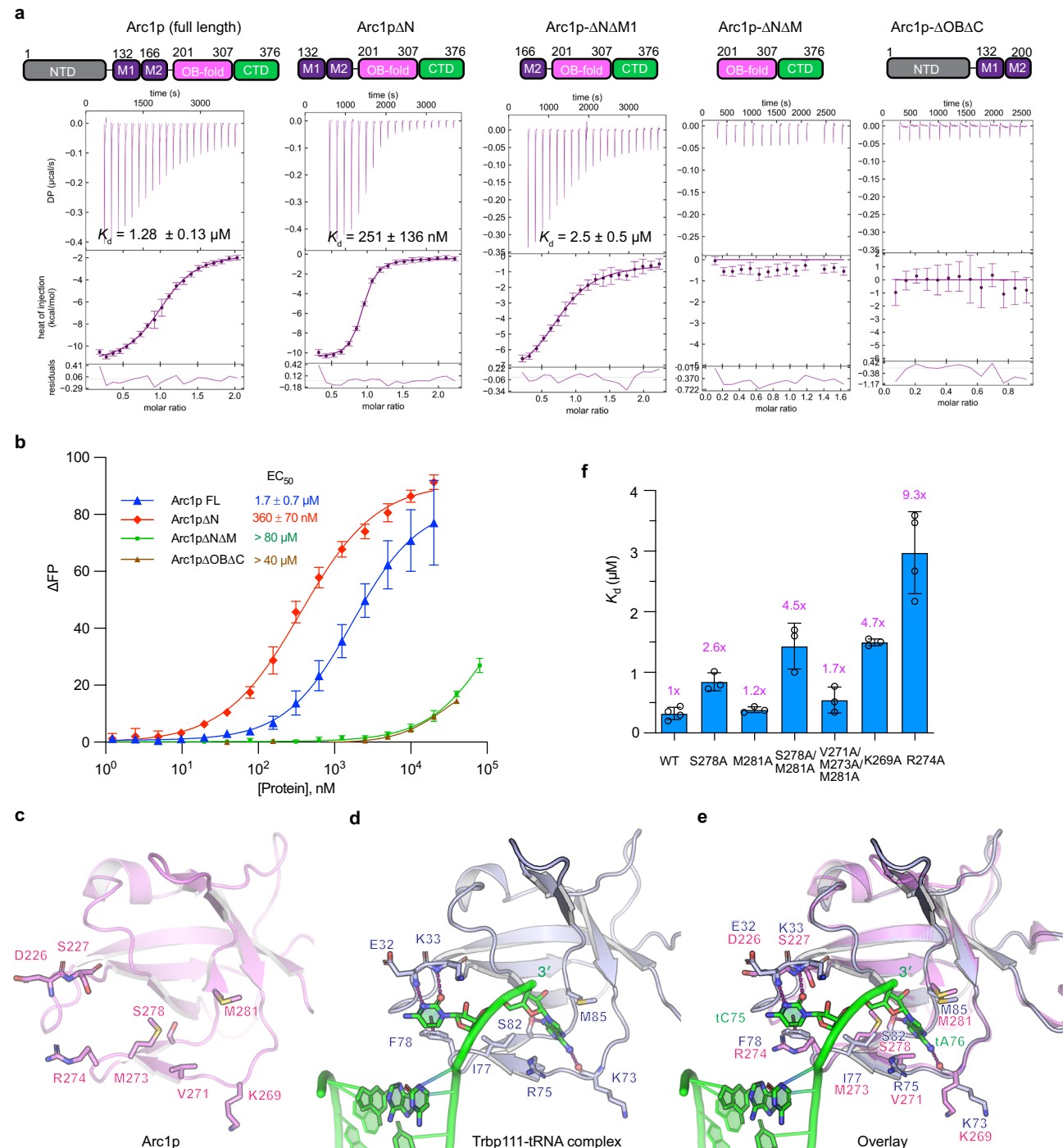

**Fig. 4 | Effects of Arc1p domain truncations and side chain substitutions on tRNA binding. a, b**, ITC (**a**) and FP (**b**) analysis of Arc1p constructs binding to *E. coli* tRNA[Ile]. **c**−**e** Structural alignment of the Arc1p OB-fold with Trbp111 in complex with tRNA showing the crystal structure of Arc1p OB domain (**c**), Trbp111-tRNA complex (**d**), and their overlay (**e**). **f** Effects of Arc1pΔN amino acid substitutions on $K_d$ measurements of tRNA[Ile] binding by ITC. Fold-change values relative to WT Arc1pΔN are indicated. Data are mean ± s.d. from n biologically independent replicates. $n = 4$ for WT, R274A; $n = 3$ for S278A, M281A, S278A/M281A, V271A/M273A/M281A, K269A.

The high degree of structural homology between the OB folds of Trbp111 and Arc1p and the evolutionary conservation of the tRNA-binding residues they share suggest a common mode of tRNA binding (Supplementary Fig. 10). Consistent with this notion, wildtype Arc1pΔN drastically increased the fluorescence lifetime of 2AP incorporated at either tC75 or tA76 position reaching ~6.2 ns, while the K269A mutant did so to much lesser extents (5.3−5.7 ns; Fig. 6a−d, Supplementary Fig. 8 and Table 2). This result demonstrates that Arc1p

directly engages the tRNA 3′ end similarly to Trbp111. In addition, deleting the 3′ ACCA from tRNA[Ile] reduced Arc1p binding by ~6.5-fold, with tA76 alone contributing 3.2-fold (Fig. 5 and Supplementary Table 4). Similarly, sequestering the tRNA tails using palindromic, dimer-forming sequences (5′ GC or 5′ GGCC) reduced binding by 5-10 folds (Supplementary Fig. 2 and Table 4). Despite the clear preference for ssRNA overhangs, Arc1p is more tolerant of sequence changes in the tail than Trbp111. Changing the terminal ACCA sequence to UGGU

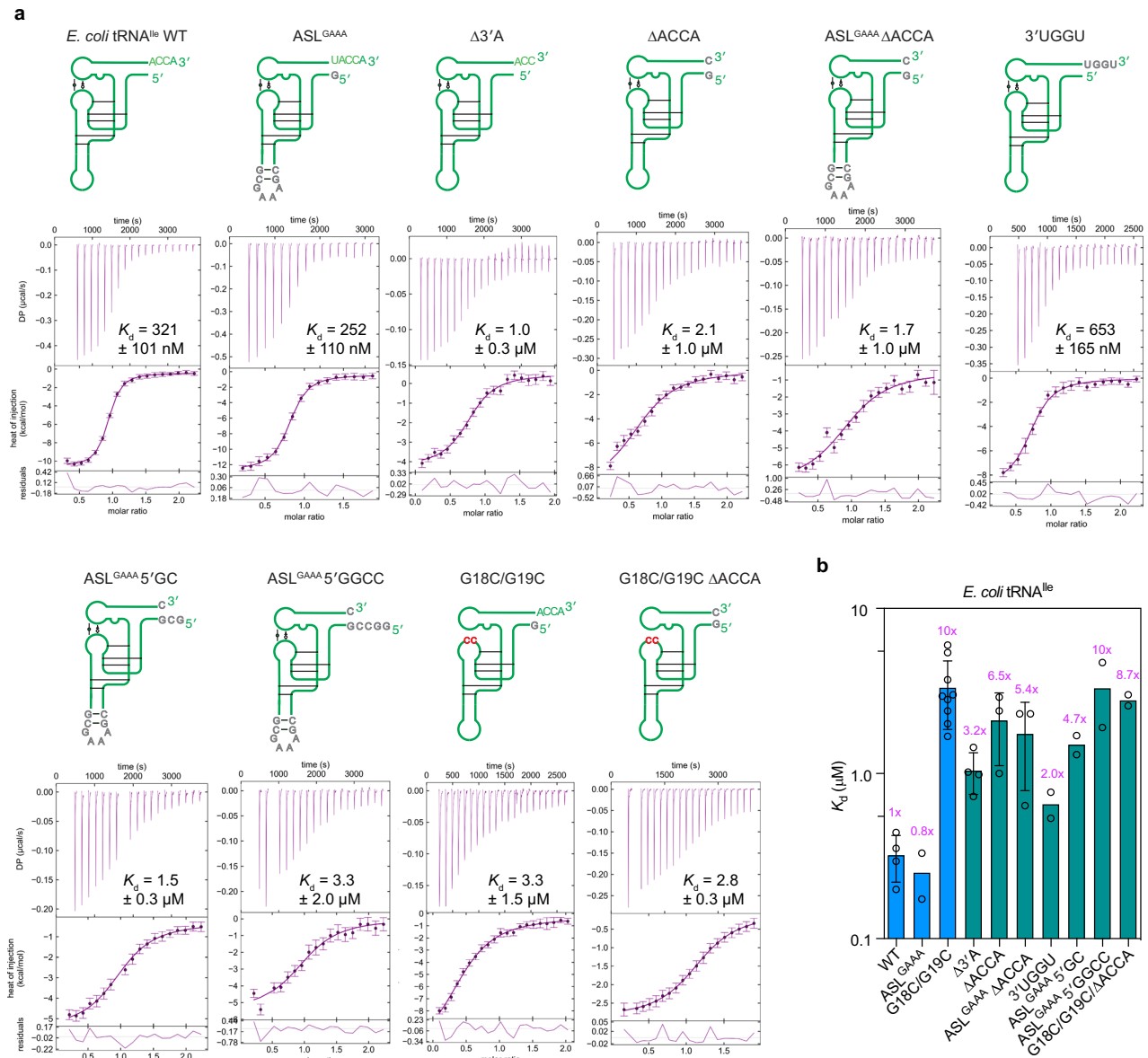

**Fig. 5 | Effects of targeted alterations to the full-length tRNA on Arc1p binding. a** Representative ITC thermograms of WT Arc1pΔN titrations with indicated tRNA$^{Ile}$ constructs. The error bars indicate uncertainties in individual injection heats estimated by NITPIC[76] (Methods). **b** Average $K_d$ values (mean ± s.d. from n biologically

independent replicates) from ITC measurements. $n = 4$ for WT, Δ3′A; $n = 2$ for ASL$^{GAAA}$, 3′UGGU, ASL$^{GAAA}$/5′GC, ASL$^{GAAA}$/5′GGCC, G18C/G19C/ΔACCA; $n = 3$ for ΔACCA, ASL$^{GAAA}$/ΔACCA; $n = 9$ for G18C/G19C. Fold-change values relative to WT tRNA are indicated.

abolished Trbp111 binding but only had a two-fold effect on Arc1p binding. Highly congruent results were also obtained using minihelices and microhelices in an FP competition assay (Fig. 6e–h and Supplementary Fig. 11). Removing the 3′ ACCA from the microhelix reduced binding by ~11 fold, while adding a U-tail also led to a 3-fold defect. Mutating tA76 to pyrimidines reduced affinity by 3-fold, while its mutation to G only resulted in a two-fold defect, consistent with the nature of the tA76 recognition. By contrast, mutation of tC75 did not substantially affect binding, nor did swapping the positions of tA76 and tC75 (MiniH-ACAC) (Fig. 6e, f). Together, these data suggest that recognition of the mature tRNA 3′ end is a major characteristic of Arc1p, but binding is more permissive of the tail sequence. This further hints at the existence of a secondary contact between Arc1p and tRNA.

## Arc1p also recognizes the tRNA body

The observation that the ΔACCA construct still robustly bound Arc1p strongly suggests that Arc1p makes additional contacts with tRNA

besides the 3′ end. In support, AUC analyses further indicated that two Arc1p molecules can simultaneously bind the same tRNA while exhibiting no tendency to dimerize on its own (Supplementary Fig. 12). We then sought to identify which part of the tRNA contributes to this secondary interaction. Like Trbp111, changing the ASL region had no impact on tRNA binding by Arc1p (Fig. 5), indicating ASL's non-involvement. By contrast, disrupting the tRNA elbow by a tG18C/tG19C substitution reduced binding by 10-fold, a finding supported both by ITC and competitive FP analyses using both full-length and ΔN Arc1p (Supplementary Fig. 13). Minihelices and microhelices also bound Arc1p with ~10-fold reduced affinity, suggesting the need for the tRNA global structure (Fig. 6). To test if the pentanucleotide T-loop motif contributes to binding, we either substituted it with a GAAA tetraloop or polyU loop or deleted it outright (Δloop) from the minihelices or microhelices, none of which strongly affected binding. To ask if Arc1p measures distances between the elbow and 3′ end as do tRNA-processing enzymes such

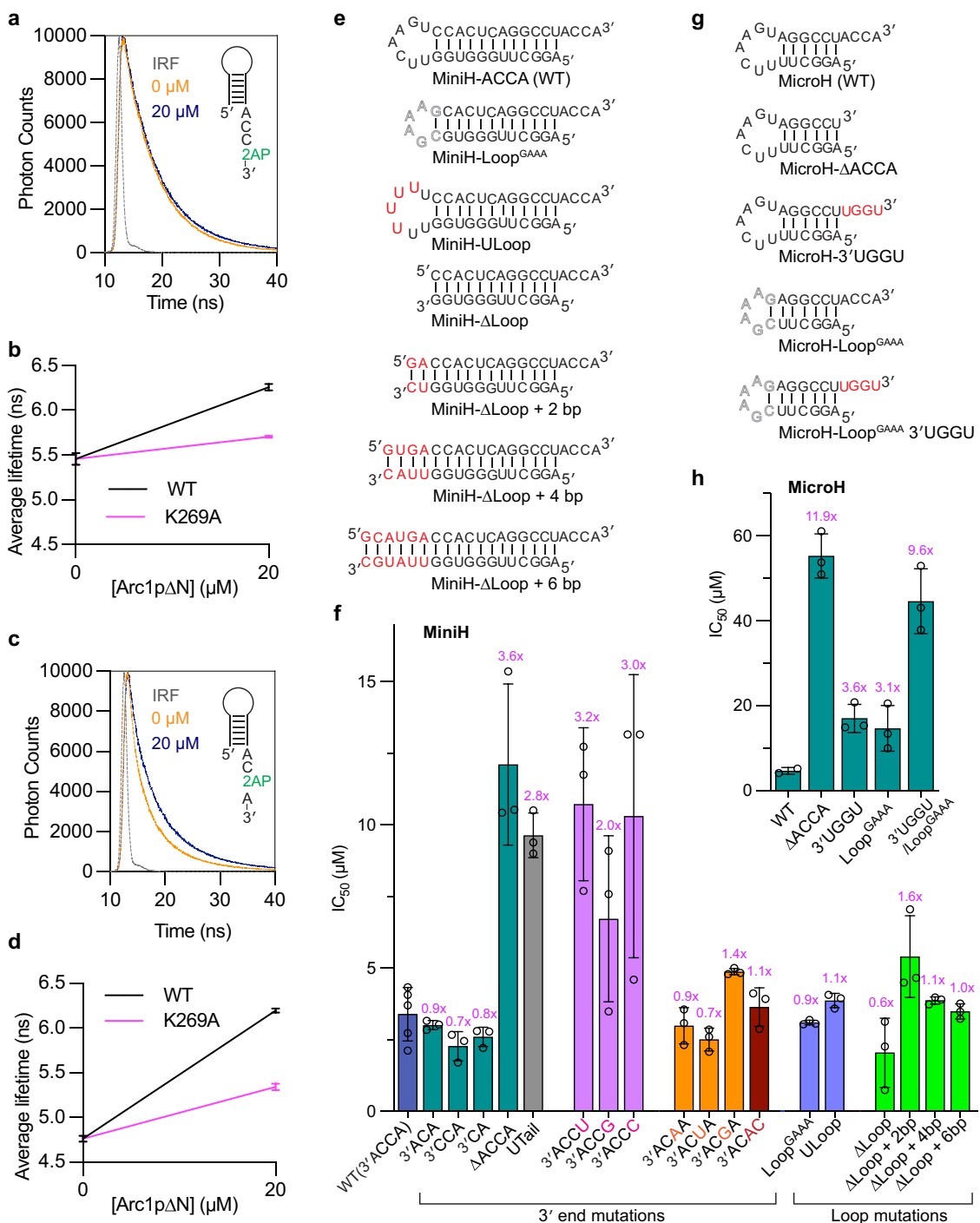

**Fig. 6 | Arc1p interactions with minihelices, microhelices, and ssRNAs. a–d.** Effects of Arc1pΔN on fluorescence lifetime of 2AP at minihelix position tA76 (MiniH-ACC-2AP) or tC75 (MiniH-AC-2AP-A). **a** Representative MiniH-ACC-2AP decay curves in the absence or presence of 20 μM Arc1pΔN. **b** Average MiniH-ACC-2AP lifetimes in the presence of wildtype or K269A Arc1pΔN. **c** Representative MiniH-AC-2AP-A decay curves in the absence or presence of 20 μM Arc1pΔN. **d** Average MiniH-AC-2AP-A lifetimes in the presence of wildtype or K269A Arc1pΔN. IRF: instrument response function. **e** Secondary structures of minihelix (MiniH) constructs used in competition FP binding assays with Arc1pΔN. **f** IC$_{50}$ values of oligos in (**e**). **g** Secondary structure of microhelix (MicroH) constructs used in competition FP binding assays with Arc1pΔN. **h** IC$_{50}$ values of oligos in (**g**). Data are mean ± s.d. from 3 biologically independent replicates. Fold-change values relative to wildtype minihelix or microhelix are indicated. See Supplementary Fig. 11 for FP titrations.

as RNase P and RNase Z, we changed the stem length of the minihelix, which also only had negligible effects on binding (Fig. 6). Interestingly, the quasi-tRNA-like human mascRNA[58] bound Arc1p with tRNA-like high affinity (Supplementary Fig. 13). Together, these data suggest that the intact tRNA fold as well as its 3′ end are required for high-affinity (nM) binding by Arc1p. To explore how the Arc1p middle

domain may recognize the tRNA body, we overlaid the AlphaFold2 model of full-length Arc1p, as a rigid body, onto our Trbp111-tRNA complex structure (Supplementary Fig. 14a). The N-terminal portion of the middle domain (M1) is placed in proximity to the tRNA core, providing a plausible overall topology for bivalent Arc1p-tRNA interaction.

## Discussion

The main findings of this work are: (1) Contrasting previous models, Trbp111 recognizes exclusively the universal 3′ end of mature tRNAs. (2) Using its OB-fold, Trbp111 precisely recognizes the single-stranded structure, 3′ terminal location, and specific sequence of the 3′ CA dinucleotide. (3) Arc1p binds tRNA 3′ end similarly as Trbp but also engages the tRNA body, likely via its basic middle domain[29–31]. This secondary contact may have shifted or otherwise constrained its OB-tRNA 3′ end interaction giving rise to the modest differences in the effects of tRNA-binding site mutations. Given the strong structural and sequence conservation of the tRNA-binding regions (Supplementary Fig. 10), the observed Trbp111-tRNA interface likely represents a general mechanism of tRNA recognition by the OB-fold.

The ancient OB-fold is an evolutionarily conserved, highly adaptable protein superfold that has diverged into 16 superfamilies to recognize different ligands from sugars to nucleic acids[59]. In *S. cerevisiae*, there are at least 18 proteins that are predicted by AlphaFold to contain the OB-fold (Supplementary Table 5). Interestingly, two structurally distinct subclasses of OB folds recognize different portions of tRNAs (Fig. 7). A previously characterized subclass of OB folds is integral to the aaRS architecture and binds 6-7 nucleotides of ASL using a wide cavity between $L_{12}$ and $L_{45}$ loops emanating from the OB barrel (Fig. 7d–f). By contrast, the Trbp/EMAPII-like OB folds employ an extended $L_{12}$ loop producing a much narrower crevice (Fig. 7a–c). The spatial juxtaposition of $L_{12}$ and $L_{45}$ allows them to jointly recognize the contiguous tC75-tA76 dinucleotide and partially blocks access to the front face of the barrel (Fig. 7b, c) Therefore, tRNA 3′ end can only approach the OB-fold from the back of the barrel, a binding mode distinctly different from most other OB folds that engage their substrates using its front face, including the ASL-binding OBs of aaRSs. For comparison, ssDNA-binding OB-folds extend the flanking β1 and β2 strands instead of the intervening $L_{12}$ loop, creating a cavity that nonspecifically clamps the ssDNA (Fig. 7g–i). This clamping can be further concatenated over an extended distance by threading long stretches of DNA through tandem or oligomeric OB-folds[60].

The binding of Trbp111 to the 3′ end of tRNA has important implications for its potential functions. Based on proposed elbow binding[12,26], Trbp111/EMAPII-like proteins were suggested to potentially have facilitated the evolution of the iconic L-shaped tRNA structure by promoting the conjugation of the two separate tRNA domains at the elbow[12]. In light of our new findings, Trbp111 may chaperone tRNA folding and prevent their entanglement or aggregation via non-native interactions by sequestering their ssRNA tails from pairing with complementary (UGGN) sequences on the tRNAs and other RNAs. Indeed, other proteins that bind tRNA 3′ ends, such as La proteins, can serve as tRNA-folding chaperones[61], suggesting that 3′ end-binding is an effective mechanism to facilitate tRNA folding. CsaA proteins that are close structural analogs of Trbp were suggested to bind tRNAs[27] and may play dual roles of protein and RNA folding chaperones (Supplementary Fig. S14b). By encasing the 3′ end in a hydrophobic environment, Trbp111 may also protect it from exonuclease degradation or chemical insults, especially in a primordial environment with elevated temperature or pH. MALAT1 and NEAT1 long noncoding RNAs protect their mature 3′ ends from degradation by concealing them within their RNA triplex cores[62,63]. Since Trbp111 tolerates some chemical adducts on the tRNA terminal ribose, it could conceivably accommodate and potentially protect the aminoacyl group from hydrolysis. If true, Trbp111 may facilitate the trafficking of aminoacyl-tRNAs to the ribosomes, or the exit and recycling of deacylated tRNAs, or otherwise function in tRNA localization control.

The frequent fusion of the Trbp/EMAPII-like OB-fold to other functional protein domains provides additional insights into the specific roles of their OB folds, such as aminoacylation, localization, and trafficking. Fusion to aaRSs has been shown to enhance the loading of tRNAs onto the enzymes by raising the local tRNA concentrations and reducing the $K_m s$[37]. The modest affinity of these OB folds further permits their subsequent handover to the catalytic domains, and may also facilitate the exit and turnover of aminoacyl-tRNAs. Within the larger MSCs, Arc1p and AIMP1 seem to serve a gatekeeper function, by facilitating and controlling the entry of tRNA substrates using their OB folds. In this context, the presence of a secondary tRNA binding site in the middle domain of both proteins closer to the NTD-anchored aaRSs[28] may help relay the tRNA substrates while maintaining contacts. Besides enhancing and gatekeeping aminoacylation, it appears that another conserved function of Trbp/EMAPII-like OB domains is to control tRNA localization and trafficking. Arc1p counters the retrograde nuclear transport of tRNAs[64], retaining them in the cytoplasm for translation. *Plasmodium* hijacks human tRNAs and imports them for translation or signaling using the membrane-anchored OB-fold of tRip[32]. Interestingly, in a role reversal, cellular tRNAs can also control the localization and trafficking of proteins, as exemplified by tRNA control of HIV-1 Gag migration toward the plasma membrane via direct interactions to regulate virion biogenesis[3].

The elucidation of the unexpected tRNA-recognition mode of the widely adopted Trbp/EMAPII-like OB-fold informs their cellular functions and mechanisms of action. It also provides a framework to understand the molecular architecture of the MSCs organized around such domains, and tRNA recruitment into and trafficking through these elaborate tRNA metabolic factories[65]. Despite their compact fold, these structurally robust, free-standing OB folds effectively select tRNAs over other competing nucleic acids. Conceivably, they could be harnessed as modular devices that target, track, or manipulate tRNAs in cells to modulate protein synthesis or tRNA-associated pathways or exert control over the expanding host and viral tRNA interactome[6].

## Methods

### Protein expression and purification

*Aquifex aeolicus* Trbp111 was expressed in *E. coli* BL21 (DE3) cells using a pD441-HMBP vector (ATUM). Cells were grown to $OD_{600}$ ~ 1 at 37 °C in a terrific broth medium containing 50 μg/mL kanamycin. Protein expression was induced with 0.5 mM isopropyl 1-thio-β-D-galactopyranoside (IPTG), and cells were grown at 20 °C overnight. Cells were harvested, resuspended in 25 mM Tris-HCl pH 7.4, 200 mM NaCl, 5% glycerol, 10 mM 2-mercaptoethanol, and stored at − 80 °C. Thawed pellets were supplemented with 1 mM phenylmethylsulfonyl fluoride (PMSF) and lyzed using a microfluidizer device. The lysate was clarified by centrifugation at $60,000 \times g$ for 45 min at 4 °C. The pellet was discarded, and the supernatant was supplemented with 15 mM imidazole, filtered, and loaded onto a HisTrap $Ni^{2+}$ column on an Akta Pure chromatography system at 4 °C. The column was washed with buffer consisting of 25 mM Tris-HCl pH 7.4, 500 mM NaCl, 15 mM imidazole, and 10 mM 2-mercaptoethanol. Bound protein was eluted in the same buffer containing 250 mM imidazole. The His-MBP tag was removed by incubating the eluted protein with PreScission protease at a 1:150 protease:protein mass ratio. Tag cleavage was carried out at 4 °C with overnight dialysis in 25 mM Tris-HCl pH 7.4, 150 mM NaCl, 2 mM $MgCl_2$. The protein mixture was then re-loaded onto a HisTrap $Ni^{2+}$ column to remove the tag and uncleaved His-MBP-Trbp111 fusion protein. The flow-through fractions were concentrated and further purified by size exclusion chromatography on a Superdex 200 column equilibrated in 25 mM Tris-HCl pH 7.4, 150 mM NaCl, and 2 mM $MgCl_2$. Plasmids for all Trbp111 mutants were generated using the QuikChange Lightning Site-directed Mutagenesis Kit (Agilent), and mutant proteins were expressed and purified following the same procedure described above. The purity and identity of the proteins were assessed and verified by SDS-PAGE and LC-MS. *Saccharomyces cerevisiae* Arc1p and Arc1p variants were expressed and purified following the same procedure outlined above, except that induction was carried out using 1 mM IPTG.

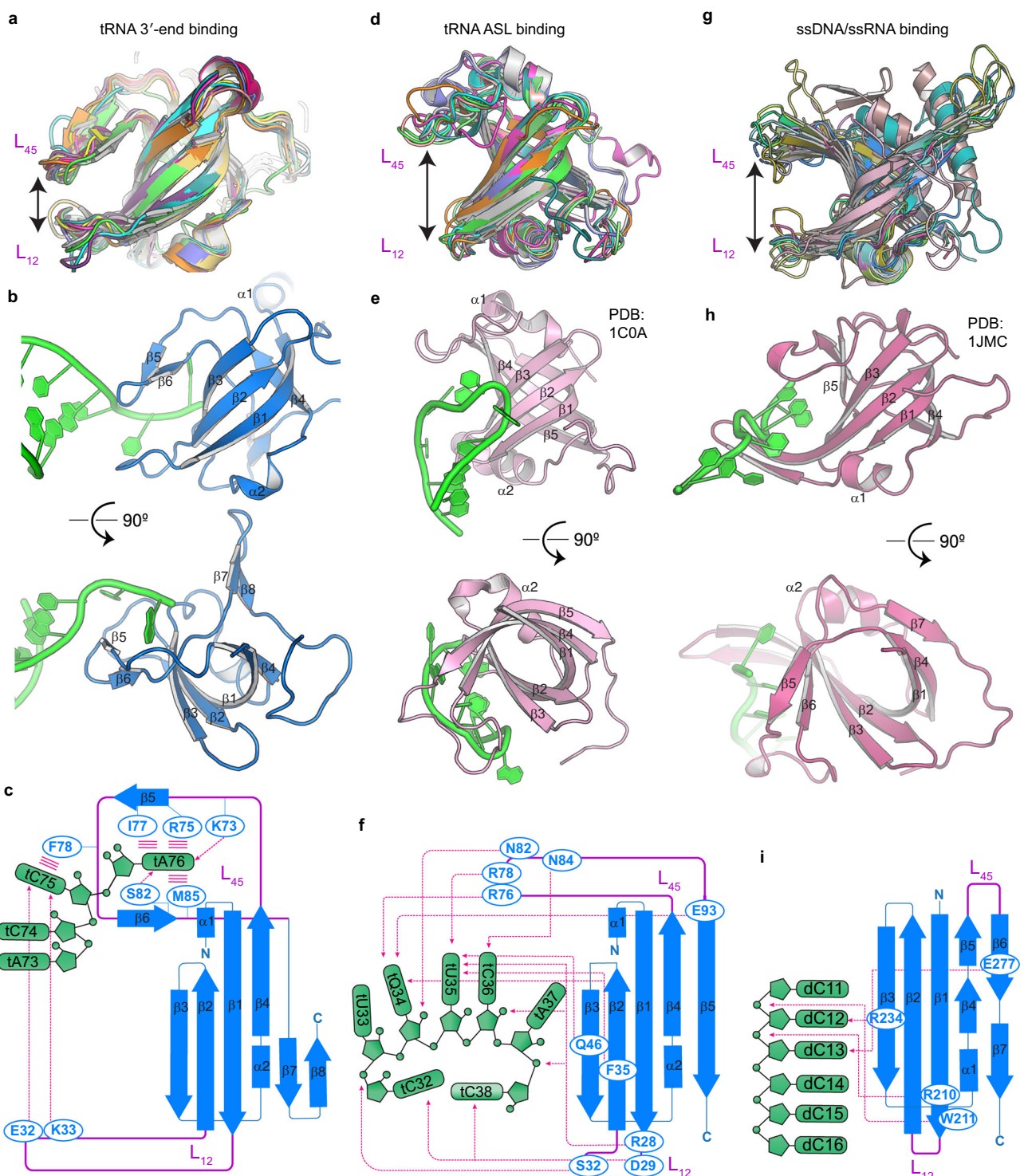

**Fig. 7 | Structural comparison of three nucleic acid-binding OB-fold classes.**
**a** Overlay of thirteen Trbp/EMAPII-like OB folds, including Trbp111. **b** Approach of tRNA to the Trbp111 OB-fold β barrel is primarily from the back side. **c** Schematic illustration of the Trbp111-tRNA interactions. All interfacial residues are located in the $L_{12}$ or $L_{45}$ loops. **d** Overlay of nine anticodon-binding OB folds. **e** Interaction of an anticodon loop with *E. coli* Aspartyl-tRNA synthetase (AspRS, PDB: 1C0A)[78]. The anticodon approaches primarily towards the front of the barrel. **f** Schematic illustration of the detailed interaction in (**e**). Residues in loops and on the barrel are utilized. **g** Overlay of fifteen ssDNA-binding OB folds. **h** ssDNA complex of one OB-fold domain of human replication protein A subunit RPA70 (PDB: 1JMC)[79]. **i** Schematic illustration of the detailed interaction ins (**h**). All contacts here involve barrel residues but no loop residues.

## RNA preparation

tRNAs used for crystallization and biophysical analyses were prepared by T7 RNA polymerase in vitro transcription using PCR-generated templates[66–68]. To reduce the N + 1 extension activity of T7 RNAP, two consecutive 2′-O-methyl modifications were introduced on the 5′ ends of the template strands. RNAs were purified by denaturing gel electrophoresis in Tris-borate EDTA buffer on 10% polyacrylamide (29:1 acrylamide:bisacrylamide) gels containing 8 M urea. RNAs were eluted by overnight crush and soak in 300 mM NaOAc pH 5.2, 1 mM EDTA at 4 °C, washed with 1 M KCl using Amicon centrifugation filters, followed

by two washes with DEPC-treated water. The concentrated RNAs were filtered and stored at −20 °C until use. Oligos used for competitive fluorescence polarization assays were purchased from Integrated DNA Technologies as lyophilized solids and used without further purification.

### Crystallization and structure determination of the Trbp111-tRNA complex

To facilitate crystallization, the anticodon loop of *E. coli* tRNA$^{Ile}$ was replaced with a stable GAAA tetraloop. This tRNA construct was folded by heating at 90 °C for 3 min in 25 mM Tris-HCl pH 7.4, 150 mM NaCl, snap-cooling on ice, then supplemented with 2 mM MgCl$_2$. RNA was then concentrated and exchanged into a low-salt buffer composed of 25 mM Tris-HCl pH 8, 50 mM KCl, and 2 mM MgCl$_2$. For crystallization, Trbp111 dimer (160 μM) was combined with 230 μM *E. coli* tRNA$^{Ile-GAAA}$ in 25 mM Tris-HCl pH 8, 50 mM KCl, 2 mM MgCl$_2$ and the complex was kept on ice for 30 min. 1 mM spermine was added and the complex was mixed 1:1 with a reservoir solution containing 0.2 M ammonium phosphate dibasic and 23% PEG 3350. Plate-shaped crystals grew over 3 days. Crystals were cryoprotected in a solution consisting of 0.2 M ammonium phosphate dibasic, 25% PEG 3350, and 30% ethylene glycol. Diffraction data were collected at the SER-CAT beamline 22-ID at the advanced photon source (APS). The crystals exhibited symmetry of space group C2$_1$. The structure was solved by molecular replacement using Phaser[69] and models of Trbp111 (PDB: 1PYB)[12] and tRNA (PDB: 6UFM)[70], resulting in a translation function Z score (TFZ) and log-likelihood gain (LLG) of 25.3 and 1208.9, respectively. The model was rebuilt in Coot[71] and refined using Phenix.Refine[72].

### Crystallization and structure determination of unbound Trbp111

During crystallization trials for the Trbp111-tRNA complex, some drops produced crystals of unbound Trbp111. Apo Trbp111 crystallized from a mixture containing 210 μM of an *E. coli* tRNA$^{Ile}$ construct and 210 μM Trbp111 dimer in 25 mM Tris-HCl pH 7.4, 150 mM NaCl, 2 mM MgCl$_2$. This equimolar mixture was combined 1:1 with a reservoir solution containing 0.2 M lithium sulfate monohydrate, 0.1 M Tris-HCl pH 8.5, and 25% PEG 3350. Needle-shaped crystals grew over 1 day. Crystals were cryoprotected in a solution consisting of 0.2 M lithium sulfate monohydrate, 0.1 M Tris-HCl pH 8.5, 25% PEG 3350, and 30 % ethylene glycol. The crystals exhibited symmetry of space group C222$_1$. The structure was solved by molecular replacement using Phaser[69] and a model of Trbp111 extracted from the tRNA complex structure, resulting in a translation function Z score (TFZ) and log-likelihood gain (LLG) of 29.7 and 950.1, respectively. The model was rebuilt in Coot[71] and refined using Phenix.Refine[72].

### Sedimentation velocity analytical ultracentrifugation (AUC)

Sedimentation velocity experiments were carried out at 50,000 rpm (201,600 x g at 7.20 cm) on a Beckman Coulter ProteomeLab XL-I analytical ultracentrifuge and An50-Ti rotor following standard protocols[73]. Stock solutions of tRNA$^{Ile-GAAA}$ and Trbp111 in 25 mM Tris-HCl pH 7.4, 50 mM NaCl, and 2 mM MgCl$_2$ were diluted in the same buffer to prepare solutions for analysis at 10 °C. Stock solutions of tRNA$^{Tyr}$ and Arc1p were prepared in low-salt (25 mM Tris-HCl pH 7.4, 50 mM NaCl, and 2 mM MgCl$_2$) or mid-salt buffer (25 mM Tris-HCl pH 7.4, 150 mM NaCl and 2 mM MgCl$_2$). Experiments were also conducted in high-salt buffer (25 mM Tris-HCl pH 7.4, 500 mM NaCl, and 2 mM MgCl$_2$). Samples for analytical ultracentrifugation were prepared in these buffers for analysis at 25 °C. Sedimentation data were analyzed in SEDFIT[74] in terms of a continuous c(*s*) distribution of sedimenting species. The solution density, viscosity, and protein partial specific volumes were calculated based on their composition in SEDNTERP[75]. A partial specific volume of 0.52 cm$^3$g$^{-1}$ was used for tRNA at 25 °C. At 10 °C, a value of 0.503 cm$^3$g$^{-1}$ was used based on initial sedimentation equilibrium experiments. Additivity rules were used to determine the partial specific volumes for the complexes. Because of differences in the partial specific volumes, experimental sedimentation coefficients present the c(*s*) distributions.

### Fluorescence polarization (FP) assay

Protein samples were exchanged from the storage buffer into the assay buffers by ultrafiltration immediately before the assay. For Arc1p, assays were run in 25 mM Tris-HCl pH 7.5, 150 mM NaCl, and 2 mM MgCl$_2$. For Trbp111, we used the same buffer but with 50 mM NaCl due to the weaker binding affinity compared to Arc1p. A working 2X solution containing 10 nM fluorescein-labeled *E. coli* tRNA$^{Tyr}$ was prepared in a buffer and kept on ice. Protein samples to be evaluated were serially diluted in the assay buffer, and 15 μL of each solution were combined with 15 μL of 2X fluorescein-labeled tRNA solution in Corning® 96-well Half Area Black Flat Bottom plates. The plate was incubated at room temperature for 5–10 min on an orbital shaker at 70 rpm. Fluorescence measurements were taken on a CLARIOstar® Plus plate reader, with excitation at 480 (16) nm and emission at 530 (40) nm. Data were analyzed in GraphPad Prism using the agonist vs. response–variable slope equation.

### Competition fluorescence polarization (FP) assay

All competition fluorescence assays were performed using *E. coli* tRNA$^{Tyr}$ labeled with fluorescein at the 3′ end. A working 2X complex solution containing 10 nM fluorescein-labeled tRNA and protein was prepared and kept on ice. Buffers used were as follows: 25 mM Tris-HCl pH 7.5, 150 mM NaCl, 2 mM MgCl$_2$ for Arc1pΔN, and 25 mM Tris-HCl pH 7.5, 50 mM NaCl, 2 mM MgCl$_2$ for Trbp111. Competing RNAs were refolded, concentrated and exchanged into assay buffer by ultrafiltration. The RNAs were then serially diluted, and 15 μL of each solution was combined with 15 μL of complex solution in a Corning® 96-well Half Area Black Flat Bottom plate. The final concentration in the assay plate of fluorescein-labeled tRNA$^{Tyr}$ in all cases was 5 nM. Concentrations of Trbp111 and full-length Arc1p were 0.75 and 1.5 μM, respectively. For Arc1pΔN, assays using full-length tRNAs contained 300 nM Arc1pΔN, while for oligos, 150 nM was used. The plate was incubated at room temperature for 10 min on an orbital shaker at 70 rpm. Fluorescence measurements were taken on a CLARIOstar® Plus plate reader, with excitation at 480 (16) nm and emission at 530 (40) nm. Data were analyzed in GraphPad Prism using the inhibitor vs. response-variable slope equation.

### 2-aminopurine fluorescence lifetime analysis by time-correlated single-photon counting (TCSPC)

Minihelix oligonucleotides containing 2-aminopurine (2AP) at terminal (MiniH-ACC-2AP) or penultimate (MiniH-AC-2AP-A) positions were folded by heating 10 μM solutions at 90 °C for 3 min in water, snap-cooling on ice, followed by addition of 25 mM Tris pH 7.4, 50 mM NaCl, 2 mM MgCl$_2$ (for Trbp111) or 25 mM Tris pH 7.4, 150 mM NaCl, 2 mM MgCl$_2$ (for Arc1pΔN). Fluorescence decay measurements in the absence of protein were performed at 10 μM RNA. For measurements in the presence of protein, protein samples were first exchanged into the assay buffer and concentrated. An aliquot was then added to pre-folded oligonucleotide and incubated on ice for ~15 min before analysis. The final concentrations were 9 μM RNA and 20 or 60 μM protein. Notably, fluorescence lifetime is generally insensitive to the fluorophore concentration. Fluorescence decays were measured on a Horiba FluoroMax Plus spectrofluorometer equipped with a DeltaTime Time-correlated single-photon counting (TCSPC) module, and a DeltaDiode-310 pulsed LED source. 2AP emission was monitored at 370 (10) nm with a preset maximum count of 10,000 photons over a measurement range of 100 ns. The instrument response function (IRF) was measured at the excitation wavelength of 313 (10) nm. Decay curves were fit to a three-exponential equation using the DAS6 fluorescence decay analysis software (Horiba).

## Isothermal titration calorimetry

ITC experiments with Arc1p variants were carried out at 25 °C on a MicroCal iTC200 microcalorimeter in an assay buffer consisting of 25 mM Tris·HCl pH 7.4, 150 mM NaCl, 2 mM MgCl$_2$. Protein samples were extensively exchanged into the assay buffer. Variants of *E. coli* tRNA$^{Ile}$ were folded by heating 25 μM solutions at 90 °C for 3 min in 25 mM Tris pH 7.4, 150 mM NaCl, snap-cooling on ice, followed by addition of MgCl$_2$ to 2 mM. RNA samples were then extensively washed with assay buffer. Titrations were performed with 230 or 200 μM protein in the syringe and 20 μM tRNA in the cell except for *E. coli* tRNA$^{Ile}$ G18C/G19C/ΔACCA where 500 μM Arc1pΔN and 50 μM tRNA were used. Thermograms were integrated using the NITPIC software[76], and data fitting was carried out in SEDPHAT[76] using the 1:1 binding model.

## Reporting summary

Further information on research design is available in the Nature Portfolio Reporting Summary linked to this article.

## Data availability

Atomic coordinates and structure factor amplitudes for *Aquifex aeolicus* Trbp111 and its complex with *E. coli* tRNA$^{Ile}$ have been deposited at the Protein Data Bank under accession codes 8VTZ and 8VU0, respectively. Source data are provided in this study. Other atomic coordinates used include Trbp111 (PDB: 1PYB) and tRNA$^{Ile}$ (PDB: 6UFM). Source data are provided in this paper.

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

## Acknowledgements

We thank I. Skeparnias for providing human mascRNA, I. Botos for computational support, G. Piszczek and D. Wu for support in biophysical analyses, and C. Stathopoulos, C. Bou-Nader, K. Suddala, I. Skeparnias, A.-N. Shaukat, and A. Brasington for discussions. This work was supported by the Intramural Research Program of the NIH, the National Institute of Diabetes and Digestive and Kidney Diseases (NIDDK) (ZIADK075136 to J.Z.), and an NIH Deputy Director for Intramural Research (DDIR) Challenge Award to J.Z. This research used resources of the Advanced Photon Source, a U.S. Department of Energy (DOE) Office of Science user facility operated for the DOE Office of Science by Argonne National Laboratory under Contract No. DE-AC02-06CH11357.

## Author contributions

A.U.J. and J.Z. conceived and designed the work. A.U.J. prepared all RNA and protein samples and crystals, performed all biochemical and biophysical assays, collected diffraction data, determined and refined the structures. A.U.J. and J.Z. analyzed the structures. R.G. performed AUC analyses. All authors contributed to data interpretation and manuscript preparation.

## Funding

## Competing interests

The authors declare no competing interests.
