## [Peer Review File · Nature Communications]

REVIEWER COMMENTS

Reviewer #1 (Remarks to the Author):

For approximately 25 years, bacterial Trbp (tRNA binding protein), featuring the OB-fold, has been suggested to interact with the elbow region, formed by the interaction between the D-loop and T-loop, of tRNA molecules. Dimeric Trbp has been proposed to facilitate tRNA folding by binding to the elbow region and enhance tRNA aminoacylation by aminoacyl-tRNA synthetases. The OB-fold of Trbp has also been identified in yeast Arc1p (aminoacyl-tRNA synthase cofactor 1), and the OB-fold of Arc1p is also important for the efficient aminoacylation of tRNAs.

Now, in this manuscript, Juru et al. deal with the structural and functional analysis of OB-fold-containing bacterial Trbp and yeast Arc1p. The authors present the crystal structure of Trbp111 from *Aquifex aeolicus* in complex with tRNA. Surprisingly, dimeric Trbp111 does not interact with the elbow region of tRNA at all; instead, it interacts only with the 3'-single-stranded C75A76 of the CCA-end of tRNA. Detailed and extensive biochemical analysis using mutant tRNA (RNA) and Trbp111 supports the specific recognition of the 3'-C75A76 of tRNA by Trbp. The authors expand the analysis to yeast Arc1p and found that the OB-fold of Arc1p, together with the adjacent region to the OB (middle), interacts with the 3'-end of tRNA as Trbp does. It was also suggested that Arc1p interacts with the tRNA body, particularly the elbow region.

This reviewer finds this study to be intriguing, and the results are quite unexpected (surprising). The study provides advanced insights into the biological role of the OB-fold in regulating tRNA aminoacylation, translation, and tRNA localization and trafficking, thereby expanding our understanding of gene regulation mediated by the well-known RNA-binding domain.

The crystallographic analysis and biochemistry in this manuscript are solid and convincing. The manuscript is well-organized and presented. This reviewer recommends publication of this manuscript in *Nature Communications* and has just a few comments.

Minor comments:

1. Some wording in the text could be rephrased.

For example,

page6, line 151, "chiefly"

page11 line 305, "similarly"

page12 line 341, "body" In the manuscript, the definition of tRNA body is ambiguous.

Page12 line 343, "beyond"

Page362 line 362, "chief"

2. The interaction between Arc1p and tRNA. It was suggested that the intact tRNA fold and 3'CCA are required for the high affinity binding by Arc1p. Although the tRNA docking onto Arc1p is modeled in Extended Data Fig 13. More detailed description should be provided in the main text.

3. How is CsaA homologous to Trbp1? If the structure of CsaA is available, comparing the structures of CsaA and Trbp111 would be interesting, as CsaA acts as protein and RNA folding chaperones.

Reviewer #2 (Remarks to the Author):

The OB fold is part of the tRNA binding protein domain superfamily yet it has not been unambiguously demonstrated how this ubiquitous fold interacts with tRNA at the atomic level (until now). Here, Juru et al have demonstrated the long-awaited mode of tRNA recognition, through extensive biochemistry, structure, and mutational analysis studies (from the perspective of both the tRNA as well as the protein). The determined co-crystal structure, considered very high quality and high resolution for a protein-tRNA complex, provides a clear picture for how the 3' end of tRNA is the primary mode of binding for the bacterial Trbp111 and how this tRNA recognition extends to the OB-fold yeast Arc1p protein system. (conserved from bacterial to yeast). This is an outstanding contribution to the field and broader community. It is very well written, with insightful details, both the extensive data and placing the results in the context of the field. Taken together, the work presents a balanced, broad perspective of the nucleic acid properties of this ubiquitous nucleic-acid binding fold. The work certainly merits publication in a high impact journal such as Nature Communications.

Some comments for consideration:

The role of the middle domain and autoinhibition of the NTD of Arc1p is only partially addressed via mutational/binding studies. Much more work needs to be done here (but can come later). The work as presented nicely focuses on the central theme of the conserved binding properties of the OB-fold. Authors clearly show that the OB-fold of Arc1p and Trbp111 bind the same location of tRNA. In this reviewer's opinion, the structure of a Arc1p-tRNA complex is beyond the scope of this work. A structure of the Arc1p-deltaN-tRNA complex is the next logical step though, and will likely come later.

As mentioned, the work irrefutably provides evidence that the OB-folds of Trbp111 and Arc1p bind the same location. However, for Trbp111 binding to the 3'end C75 vs A76 position, the R75A (impacting A76) has a substantially bigger effect than F78A (impacting C75) (see Fig 3B). Yet, in Fig 4F, for the Arc1p system, it is the reverse. There is a substantially higher degree of binding specificity for the C75 position (R274A (impacting C75) affects binding more so than S278A (impacting A76)). This suggests that there is a difference in how Trbp111 and Arc1p discriminate the 3' end. Could authors expand upon their comments here, either in the results and/or the discussion?

Negative cooperativity is clearly observed from high quality and well analyzed AUC data. This seems like a very interesting result and is well presented in the results, yet not mentioned in the discussion. Is this a worthy result to additionally emphasize in the discussion section?

Fig 7 shows how the ancient OB fold binds tRNA (this work), the ASL of tRNA (aaRS), and ssDNA (via RPA70). An effective figure. It is mentioned in the legend, but please also include PDBs in the actual figure to emphasize all images are from experimentally determined structures. Regarding this figure and the discussion: Is there any biological evidence that RPA binds tRNA 3' ends in vivo OR that Trbp111 or Arc1p bind ssDNA in vivo. Authors mention potential cell functions regarding tRNA mechanism of action in the discussion (trafficking, recycling, potential as a chaperone) for the OB-fold. Fig 7 and these structures suggest that many OB-folds from other proteins could moonlight to impact tRNA biology, and that there may be more proteins that impact tRNA biology than is currently appreciated. Thus, as it relates to Fig7, could authors provide a comprehensive list that identifies all known yeast proteins that are predicted to contain an OB-fold? Or perhaps cite

a reference of such a study if already published? Such a bioinformatics exercise would be supplemental information and is not necessary for publication of this work, although it seems some analysis to this effect would strengthen the broad themes presented in the discussion.

Minor comments:

Lines 51/52 (remove ',etc') so it reads "WHEP domain, and Zinc-finger domains."

Line 62 (reword): and are designated as

Line 95-96 (suggestion: simplify sentence structure by numbering both functionalities): of 1) gating the entry..... and 2) harboring the latent...

Line 100 (remove ',etc')

Line 145 suggestion (replace chiefly with primarily)

Line 296 (significant figs issue), write either as 1.300 +/- 0.013 OR as 1.30 +/- 0.01

Line 347 (replace the word 'contract' with contrast)

Extended Figs 1 and 4: Electron density is shown. State the sigma levels in the figure legend. Also, what software(s) used to render the images (Pymol or Coot (both), or other software).

Extended Fig 5: How was surface area calculated (what program)? Plotted via excel (please specify visualization software)

Extended Fig 6: The protein is pre-organized as authors clearly show. However, there are minor local movements observed and there appears to be overall, "global" changes in tRNA. This figure shows an alteration of the relative positions of the anticodon stem loop. Are these changes due to a crystal packing effect or are they representative of changes in tRNA structure/flexibility due to binding? Please comment (in figure legend).

Point-by-point responses to reviewer comments:

Reviewer #1 (Remarks to the Author):

For approximately 25 years, bacterial Trbp (tRNA binding protein), featuring the OB-fold, has been suggested to interact with the elbow region, formed by the interaction between the D-loop and T-loop, of tRNA molecules. Dimeric Trbp has been proposed to facilitate tRNA folding by binding to the elbow region and enhance tRNA aminoacylation by aminoacyl-tRNA synthetases. The OB-fold of Trbp has also been identified in yeast Arc1p (aminoacyl-tRNA synthase cofactor 1), and the OB-fold of Arc1p is also important for the efficient aminoacylation of tRNAs. Now, in this manuscript, Juru et al. deal with the structural and functional analysis of OB-fold-containing bacterial Trbp and yeast Arc1p. The authors present the crystal structure of Trbp111 from *Aquifex aeolicus* in complex with tRNA. Surprisingly, dimeric Trbp111 does not interact with the elbow region of tRNA at all; instead, it interacts only with the 3'-single-stranded C75A76 of the CCA-end of tRNA. Detailed and extensive biochemical analysis using mutant tRNA (RNA) and Trbp111 supports the specific recognition of the 3'-C75A76 of tRNA by Trbp. The authors expand the analysis to yeast Arc1p and found that the OB-fold of Arc1p, together with the adjacent region to the OB (middle), interacts with the 3'-end of tRNA as Trbp does. It was also suggested that Arc1p interacts with the tRNA body, particularly the elbow region.

This reviewer finds this study to be intriguing, and the results are quite unexpected (surprising). The study provides advanced insights into the biological role of the OB-fold in regulating tRNA aminoacylation, translation, and tRNA localization and trafficking, thereby expanding our understanding of gene regulation mediated by the well-known RNA-binding domain.

The crystallographic analysis and biochemistry in this manuscript are solid and convincing. The manuscript is well-organized and presented. This reviewer recommends publication of this manuscript in *Nature Communications* and has just a few comments.

A: We thank the reviewer for their favorable assessments and helpful suggestions.

Minor comments:

1. Some wording in the text could be rephrased.

For example,

page6, line 151, “chiefly”

A: We changed it to “primarily”.

page11 line 305, “similarly”

A: We rephrased it as “Arc1p recognizes tRNA 3' end in a manner similar to Trbp111.”

page12 line 341, “body” In the manuscript, the definition of tRNA body is ambiguous.

A: We added explicit definition of the tRNA body in the Introduction. We now state:

“Arc1p similarly binds tRNA 3' end as Trbp and supplements this contact with an additional contact to the tRNA body (defined as the non-tail portion)”.

Page12 line 343, “beyond”

A: We changed it to “besides”.

Page362 line 362, “chief”

A: We changed it to “main”.

2. The interaction between Arc1p and tRNA. It was suggested that the intact tRNA fold and 3'CCA are required for the high affinity binding by Arc1p. Although the tRNA docking onto Arc1p is modeled in Extended Data Fig. 13. More detailed description should be provided in the main text.

A: We have added the description at the end of the results section. We simply overlaid the AlphaFold model on top of the Trbp111-tRNA complex structure, both as rigid bodies. We refrained from further structural manipulation or fitting, as the domain orientations predicted by AlphaFold may not be reliable. We now state:

“To explore how the Arc1p middle domain may recognize the tRNA body, we overlaid the AlphaFold2 model of full-length Arc1p, as a rigid body, onto our Trbp111-tRNA complex structure (Supplementary Fig. 14). The N-terminal portion of the middle domain (M1) is placed in proximity to the tRNA core, providing a plausible overall topology for bivalent Arc1p-tRNA interaction.”

3. How is CsaA homologous to Trbp1? If the structure of CsaA is available, comparing the structures of CsaA and Trbp111 would be interesting, as CsaA acts as protein and RNA folding chaperones.

A: Yes CsaA is structurally very similar to Trbp111 in structure. We have now added a structural comparison between the two proteins, in a new Supplementary Fig. 14 panel b, below.

Reviewer #2 (Remarks to the Author):

The OB fold is part of the tRNA binding protein domain superfamily yet it has not been unambiguously demonstrated how this ubiquitous fold interacts with tRNA at the atomic level (until now). Here, Juru et al have demonstrated the long-awaited mode of tRNA recognition, through extensive biochemistry, structure, and mutational analysis studies (from the perspective of both the tRNA as well as the protein). The determined co-crystal structure, considered very high quality and high resolution for a protein-tRNA complex, provides a clear picture for how

the 3' end of tRNA is the primary mode of binding for the bacterial Trbp111 and how this tRNA recognition extends to the OB-fold yeast Arc1p protein system. (conserved from bacterial to yeast). This is an outstanding contribution to the field and broader community. It is very well written, with insightful details, both the extensive data and placing the results in the context of the field. Taken together, the work presents a balanced, broad perspective of the nucleic acid properties of this ubiquitous nucleic-acid binding fold. The work certainly merits publication in a high impact journal such as Nature Communications.

A: We thank the reviewer for their favorable assessments and helpful suggestions.

Some comments for consideration:

The role of the middle domain and autoinhibition of the NTD of Arc1p is only partially addressed via mutational/binding studies. Much more work needs to be done here (but can come later). The work as presented nicely focuses on the central theme of the conserved binding properties of the OB-fold. Authors clearly show that the OB-fold of Arc1p and Trbp111 bind the same location of tRNA. In this reviewer's opinion, the structure of a Arc1p-tRNA complex is beyond the scope of this work. A structure of the Arc1p-deltaN-tRNA complex is the next logical step though, and will likely come later.

A: Yes. We couldn't agree more. Indeed, we have also attempted to solve the Arc1p-deltaN-tRNA complex structure as well but so far have not succeeded despite extensive trials. Efforts are currently on-going to determine this structure as a follow-up to this current study.

As mentioned, the work irrefutably provides evidence that the OB-folds of Trbp111 and Arc1p bind the same location. However, for Trbp111 binding to the 3' end C75 vs A76 position, the R75A (impacting A76) has a substantially bigger effect than F78A (impacting C75) (see Fig 3B). Yet, in Fig 4F, for the Arc1p system, it is the reverse. There is a substantially higher degree of binding specificity for the C75 position (R274A (impacting C75) affects binding more so than S278A (impacting A76)). This suggests that there is a difference in how Trbp111 and Arc1p discriminate the 3' end. Could authors expand upon their comments here, either in the results and/or the discussion?

A: We appreciate this insightful, astute observation by the reviewer. Indeed, there are some curious differences between how Trbp111 and Arc1p binds the 3' end of the tRNA, which we don't fully understand. Besides this differential specificity on C75/A76 recognition, we also noted that the standalone OB fold of Arc1p cannot bind tRNA, while Trbp111 is self-sufficient. The most likely explanation may stem from the secondary interaction between the basic middle domain and the tRNA body. For instance, the middle domain-tRNA body engagement may constrain how its tethered OB fold engages the tRNA. A minor movement of the tRNA 3' tail relatively to the OB fold, imposed by the adjacent middle domain-tRNA body contact, may alter the relative contributions of the C75 interface versus the A76 interface.

We have now highlighted this observation and added this potential explanation at the beginning of Discussion. Now we state:

"This secondary contact may have shifted or otherwise constrained its OB-tRNA 3' end interaction giving rise to the modest differences in the effects of tRNA-binding site mutations"

Negative cooperativity is clearly observed from high quality and well analyzed AUC data. This

seems like a very interesting result and is well presented in the results, yet not mentioned in the discussion. Is this a worthy result to additionally emphasize in the discussion section?

A: While we fully agree with the reviewer that the effect seems robust and clear, we don't yet have a plausible explanation for the negative cooperativity. Comparing the apo and tRNA-bound structures did not immediately reveal how tRNA binding to the first protomer negatively impact the second tRNA-binding site. Without a plausible mechanistic underpinning of the observed allostery, we have refrained from speculating too much here.

Fig 7 shows how the ancient OB fold binds tRNA (this work), the ASL of tRNA (aaRS), and ssDNA (via RPA70). An effective figure. It is mentioned in the legend, but please also include PDBs in the actual figure to emphasize all images are from experimentally determined structures. Regarding this figure and the discussion: Is there any biological evidence that RPA binds tRNA 3' ends in vivo OR that Trbp111 or Arc1p bind ssDNA in vivo. Authors mention potential cell functions regarding tRNA mechanism of action in the discussion (trafficking, recycling, potential as a chaperone) for the OB-fold. Fig 7 and these structures suggest that many OB-folds from other proteins could moonlight to impact tRNA biology, and that there may be more proteins that impact tRNA biology than is currently appreciated. Thus, as it relates to Fig7, could authors provide a comprehensive list that identifies all known yeast proteins that are predicted to contain an OB-fold? Or perhaps cite a reference of such a study if already published? Such a bioinformatics exercise would be supplemental information and is not necessary for publication of this work, although it seems some analysis to this effect would strengthen the broad themes presented in the discussion.

A: As suggested we have now embedded the PDB codes in the actual figures (panels e and h). As far as we know, there are no reported evidence of cross-reactivity among these different OB subclasses that bind ssDNA, tRNA ASL, and tRNA 3' end, etc. Given the substantial structural differences in their L12 and L45 protrusions, we think it more likely that their substrate specificities have sufficiently diverged so as not to cross-react. However, it is a worthwhile experiment to experimentally clarify this in a follow-up study.

As to a comprehensive list of all OB-containing proteins in yeast, we obtained a list of 18 proteins that are predicted by AlphaFold to contain an OB fold, using the Foldseek server (van Kempen M et al., 2023, *Nat. Biotech*; <https://search.foldseek.com/search>). We have now included this information below, and as a new Supplementary Table 5. Arc1p is featured prominently at the top, as expected, and we inspected several other entries on the list, and verified the presence of an OB fold in their predicted structures. In the Discussion, we have added the following text:

“In S. cerevisiae, there are at least 18 proteins that are predicted by AlphaFold to contain the OB fold (Supplementary Table 5).”

Target	Description	Prob.	E-Value	Score	Query Pos.	Target Pos.
AF-P46672-F1-model v4	tRNA-aminoacylation cofactor ARC1	1	4.40E-05	217	12-105 (107)	210-305 (376)
AF-P53732-F1-model v4	37S ribosomal protein S12, mitochondrial	0.25	8.37E-01	42	1-107 (107)	45-150 (153)
AF-Q02950-F1-model v4	37S ribosomal protein MRP51, mitochondrial	0.16	1.12E+00	37	13-98 (107)	226-302 (344)
AF-P38861-F1-model v4	60S ribosomal export protein NMD3	0.12	4.32E+00	33	12-101 (107)	314-408 (518)
AF-P04802-F1-model v4	Aspartate--tRNA ligase, cytoplasmic	0.12	2.87E+00	33	1-95 (107)	94-198 (557)
AF-Q3E7X9-F1-model v4	40S ribosomal protein S28-A	0.1	5.16E+00	31	10-69 (107)	4-61 (67)
AF-P33299-F1-model v4	26S proteasome regulatory subunit 7 homolog	0.1	3.63E+00	31	2-83 (107)	84-187 (467)
AF-Q08162-F1-model v4	Exosome complex exonuclease DIS3	0.1	3.63E+00	31	1-65 (107)	260-319 (1001)
AF-P24384-F1-model v4	Pre-mRNA-splicing factor ATP-dependent RNA helicase PRP22	0.1	1.69E+00	31	1-107 (107)	153-261 (1145)
AF-POCX30-F1-model v4	40S ribosomal protein S23-B	0.09	1.59E+00	30	12-93 (107)	47-124 (145)
AF-Q01939-F1-model v4	26S proteasome regulatory subunit 8 homolog	0.09	3.63E+00	30	5-83 (107)	58-126 (405)
AF-POCX29-F1-model v4	40S ribosomal protein S23-A	0.08	1.90E+00	29	12-93 (107)	47-124 (145)
AF-P26754-F1-model v4	Replication factor A protein 2	0.08	6.91E+00	29	13-98 (107)	70-159 (273)
AF-P20459-F1-model v4	Eukaryotic translation initiation factor 2 subunit alpha	0.08	3.42E+00	29	12-92 (107)	18-86 (304)
AF-P38859-F1-model v4	DNA replication ATP-dependent helicase/nuclease DNA2	0.08	5.16E+00	29	3-82 (107)	397-500 (1522)
AF-POCX0-F1-model v4	40S ribosomal protein S28-B	0.07	9.27E+00	27	10-69 (107)	4-61 (67)
AF-P20436-F1-model v4	DNA-directed RNA polymerases I, II, and III subunit RPABC3	0.06	8.74E+00	25	11-72 (107)	5-69 (146)
AF-P29366-F1-model v4	Bud emergence protein 1	0.03	6.52E+00	21	1-95 (107)	145-208 (551)

Minor comments:

Lines 51/52 (remove ‘,etc’) so it reads “WHEP domain, and Zinc-finger domains.”

A: Revised as suggested.

Line 62 (reword): and are designated as

A: Revised as suggested.

Line 95-96 (suggestion: simplify sentence structure by numbering both functionalities): of 1) gating the entry..... and 2) harboring the latent...

A: Revised as suggested.

Line 100 (remove 'etc')

A: Revised as suggested.

Line 145 suggestion (replace chiefly with primarily)

A: Revised as suggested.

Line 296 (significant figs issue), write either as 1.300 +/- 0.013 OR as 1.30 +/- 0.01

A: Revised as suggested.

Line 347 (replace the word 'contract' with contrast)

A: Thank you for catching this! Revised as suggested.

Extended Figs 1 and 4: Electron density is shown. State the sigma levels in the figure legend. Also, what software(s) used to render the images (Pymol or Coot (both), or other software).

A: Added these missing information to the figure legend (1.0 sigma and MacPyMOL).

Extended Fig 5: How was surface area calculated (what program)? Plotted via excel (please specify visualization software)

A: Added these missing information to the figure legend (StrucTools; Excel; MacPyMOL).

Extended Fig 6: The protein is pre-organized as authors clearly show. However, there are minor local movements observed and there appears to be overall, "global" changes in tRNA. This figure shows an alteration of the relative positions of the anticodon stem loop. Are these changes due to a crystal packing effect or are they representative of changes in tRNA structure/flexibility due to binding? Please comment (in figure legend).

A: Perhaps it is the viewing angle. The bodies of the two tRNA molecules actually are quite similar (Fig. S6b), with an all-atom RMSD of 1.3 Å and C1' RMSD of 1.0 Å, including the flexing 3'-tails, without outlier rejection (using "align" function with cycles=0 in PyMOL). When outlier rejection is enabled (1272 out of 1547 atoms aligned), the tRNA-tRNA RMSD is merely 0.38 Å, suggesting the bulk of the two tRNA structures are well superimposable.

Given the average coordinate error of 0.47 Å in this structure, the minor coordinate differences between the two tRNAs appear insignificant. We have added these RMSD values to the figure legend. The local crystal-packing environment of the two engineered anticodon regions are also nearly identical (red, GAAA tetraloop engaging similar A-minor interactions, right figure).

REVIEWERS' COMMENTS

Reviewer #1 (Remarks to the Author):

The authors revised the manuscript thoroughly, and the manuscript is now acceptable for publication in Nature Communications.

Although not required for the revision, this reviewer is curious about the model construction of tRNA binding to Arc1p using AlphaFold3, which was released during the revision. Did the authors try AlphaFold3? Is it same as shown in Supplementary Figure 14?

Reviewer #2 (Remarks to the Author):

Authors have sufficiently addressed all comments from the R1 and R2 first round of reviews. They have further strengthened the manuscript, and from this reviewer's perspective, no further revisions are needed. Thank you for the opportunity to review this manuscript and this discovery is suitable for publication in a top journal such as Nature Communications.

Point-by-point responses to reviewer comments:

Reviewer #1 (Remarks to the Author):

The authors revised the manuscript thoroughly, and the manuscript is now acceptable for publication in Nature Communications.

Although not required for the revision, this reviewer is curious about the model construction of tRNA binding to Arc1p using AlphaFold3, which was released during the revision. Did the authors try AlphaFold3? Is it same as shown in Supplementary Figure 14?

A: We thank the reviewer for their favorable assessments and helpful suggestions. Yes. Out of curiosity we did try AlphaFold3 to predict tRNA binding to either Trbp111 (left) and Arc1p (middle). As shown below, AF3 seems to incorrectly predict anticodon binding by Trbp111. Interestingly, AF3 seems to correctly predict tRNA 3' end binding by Arc1p (middle), similar but not identical to the docking model in Supp Fig. 14 (right; green: Trbp111-tRNA co-crystal structure; cyan: Arc1p AF2 model; magenta: AF3 model of tRNA-Arc1p complex).

Reviewer #2 (Remarks to the Author):

Authors have sufficiently addressed all comments from the R1 and R2 first round of reviews. They have further strengthened the manuscript, and from this reviewer's perspective, no further revisions are needed. Thank you for the opportunity to review this manuscript and this discovery is suitable for publication in a top journal such as Nature Communications.

A: We thank the reviewer for their favorable assessments and helpful suggestions.